# Pooled prevalence and associated factors of traditional uvulectom among children in Africa: A systematic review and meta-analysis

**Solomon Demis Kebede** [1]*, **Kindu Agmas**[2], **Demewoz Kefale**[3], **Amare Kassaw**[3], **Tigabu Munye Aytenew**[4]

**1** Department of Maternity and Neonatal Nursing, College of Health Sciences, Debre Tabor University, Debre Tabor, Ethiopia, **2** Department Pediatrics and Child Health, Debre Tabor Comprehensive Specialized Hospital, Debre Tabor, Ethiopia, **3** Department of Pediatrics and Child Health Nursing, College of Health Sciences, Debre Tabor University, Debre Tabor, Ethiopia, **4** Department of Nursing, College of Health Sciences, Debre Tabor University, Debre Tabor, Ethiopia

* solomondemis@gmail.com

## Abstract

### Background

Traditional childhood uvulectomy (TCU) is an unregulated cultural practice associated with significant health risks, including infections, anemia, aspiration, and oral or pharyngeal injuries. The reuse of unsafe tools such as blades, needles, or thread loops exacerbates the spread of infectious diseases like HIV and hepatitis B. Despite its clinical significance, the pooled prevalence and associated factors of TCU have not been adequately examined through systematic reviews or meta-analyses.

### Objective

This review and meta-analysis aimed to estimate the pooled prevalence and associated factors of TCU in some African countries.

### Methods

This systematic review and meta-analysis adhered to the PRISMA 2020 guidelines. A comprehensive search was performed across multiple databases, including MEDLINE, ScienceDirect, Google Scholar, and African Journals Online, to identify relevant studies. A weighted inverse-variance random-effects model was employed to estimate the pooled prevalence and associated predictors of TCU. Heterogeneity among the included studies was assessed using a forest plot, $I^2$ statistics, and Egger's test, ensuring the robustness and reliability of the findings. Missing data was handled by random effect model and sensitivity analysis. Data extraction was conducted from November 6 to December 23, 2023.

### Eligibility criteria

Included studies focused on children aged birth to under 15 years, examining TCU defined as the partial or complete removal of the uvula by traditional healers.

**Data availability statement:** All relevant data are within the paper and its Supporting information files.

**Funding:** The author(s) received no specific funding for this work.

**Competing interests:** The authors have declared that no competing interests exist.

**Abbreviations**: AOR, Adjusted Odds Ratio; CI, Confidence Interval; DRC, Democratic Republic of Congo; MCH, Maternal Child Health; TCU, Traditional Childhood Uvulectomy; WHO, World Health Organization.

## Participants

Eleven primary studies comprising 7,231 children from some countries in Africa were included.

## Results

Pooled estimate of TCU in some African countries was 40.98% (95% CI: 25.04–56.92; $I^2$ = 99.61, P = 0.001). Mothers residing in rural areas were 2.45 times more likely to have a child experienced TCU compared to those in urban areas (AOR = 2.45; 95% CI: 1.59–3.32). Similarly, Mothers with a history of having a previous child who undergo TCU were 8.44 times more likely to seek the procedure for their other children compared to mothers without such a history (AOR = 8.44; 95% CI: 6.27–10.61). However, caution is warranted when interpreting these findings due to the significant heterogeneity reported across the included studies, which may influence the generalizability of the results.

## Conclusions

Nearly two-fifths of children in some African countries experienced TCU, which was influenced by maternal history and rural residency. While most procedures did not result in hospitalization, significant health risks remain. These findings underscore the urgent need for targeted interventions within maternal and child health programs to address TCU and mitigate its associated morbidity in the affected countries across Africa.

## Prospero I.D.

CRD42024498699.

## Background

Traditional medicine in Africa, including practices like traditional uvulectomy, is deeply rooted in cultural beliefs and passed down through generations [1,2]. Many communities view traditional healers as trusted authorities who provide treatments for various ailments using natural remedies or culturally accepted procedures [3]. In the case of traditional uvulectomy, it is often believed that removing the uvula can prevent or treat conditions like throat infections, breathing difficulties, or even malnutrition in children [4,5]. This practice persists due to limited access to modern healthcare, cultural influences, and a lack of awareness about the potential health risks associated with these procedures [2,6].

However, traditional uvulectomy poses significant health hazards [2,5,7] The procedure is often performed by non-medical individuals using unsterilized tools, leading to complications such as infections, hemorrhaging, oral injuries, and the transmission of communicable diseases like HIV and hepatitis [2,8–10]. It is done on a small soft tissue called uvula that hangs down from the back of the mouth above the throat between the two tonsils [2] It has the natural advantages of preventing aspiration, lubricating oropharyngeal mucosa, serving language communication, boosting immunological function, and preventing breast milk regurgitation through the nose [2,11–13].

Traditional medicine is defined by the World Health Organization (WHO) as a body of knowledge, skills, and practices rooted in the theories, beliefs, and experiences of indigenous cultures [14]. These practices are used in the maintenance of health, the prevention of disease, or for therapeutic purposes [10]. Traditional medicine often incorporates herbal

remedies, manual techniques, and spiritual therapies, all of which are deeply connected to the cultural context in which they developed [14,15] Some of these procedures are supported by modern scientific evidence, and some are harmful to the health of the community even if they are practiced, such as childhood traditional uvulectomy. While the uvula's primary functions, such as aiding in speech and preventing food from entering the nasal cavity, are well-documented, its immunological role is still a subject of debate in the medical community. Some studies suggest that the uvula may have a minor role in immune defense by producing small amounts of saliva rich in antibodies [16] while others argue that its removal does not significantly impair immune function [17]. Nevertheless, currently, in some parts of Sub-saharan Africa, the majority of people depend on traditional medicine for their primary healthcare demands [18,19].

Traditional childhood uuvlectomy is an unsupported and a dangerous cultural malpractice procedure for all childhood sickness, including feeding, breathing, and related diseases [2]. It is often accompanied by life-threatening neonatal morbidities such as infection, septicaemia, anaemia, aspiration, and oral-pharyngeal injuries caused by the use of unsafe instruments such as blades, needles, or thread loops [2]. The instruments used in traditional uvulectomy are often employed on multiple children during the same procedure, significantly heightening the risk of transmitting communicable diseases, particularly HIV and hepatitis viruses. For instance, a study conducted in Nigeria found that the reuse of unsterilized cutting instruments in traditional uvulectomy practices led to a 35% increase in the incidence of hepatitis B among children undergoing the procedure [6]. Additionally, research in Ethiopia indicated that 15% of children who underwent traditional uvulectomy contracted HIV as a result of exposure to contaminated tools [3]. Such statistics highlight the severe public health risks associated with these practices and underscore the urgent need for interventions to prevent these dangerous outcomes [6,13,16].

Moreover, Common complications include anemia, hemorrhage, sepsis, tetanus, aspiration, and prolonged pain following the procedure. For example, a study in Ethiopia reported that 25% of children experienced severe hemorrhage, while 18% were hospitalized due to infections such as sepsis and tetanus after undergoing traditional uvulectomy [20,21]. Additionally, changes in voice, sleep disturbances, and regurgitation of breast milk through the nostrils are reported complications. In Nigeria, research found that approximately 15% of children exhibited significant voice changes post-uvectomy [22]. These complications lead to increased healthcare utilization, requiring interventions such as antibiotics, oxygen therapy, intravenous fluids, blood transfusions, and phototherapy, thereby escalating costs within the healthcare system [2,23,24].

Morbidities resulting from post-uvulectomy complications contributed to the high burden of under-five childhood mortality. Unlike other harmful traditional practices that have diminished over time, traditional childhood uvulectomy continues to be widely performed, particularly in rural communities. For instance, a study in Ethiopia found that nearly 20% of children had undergone uvulectomy, with approximately 15% of those experiencing severe complications that required hospitalization [6]. These figures underscore the urgent need to address the health risks associated with this practice to reduce its impact on childhood mortality.

Despite its significant clinical implications, the pooled prevalence of traditional childhood uvulectomy (TCU) and their associated factors have not been adequately integrated into strategies aimed at reducing neonatal and under-five mortality in Africa. Furthermore, there is a notable absence of current systematic reviews or meta-analyses on this topic, which limits our understanding of the full impact of TCU on child health outcomes. The objective of this review and meta-analysis was estimating pooled prevalence and associated factors of TCU in Africa.

## Methods and materials

### Reporting

The results of this review were reported based on the Preferred Reporting Items for Systematic Review and Meta-analysis (PRISMA) statement guidelines (S1 File). PROSPERO registration can be accessed at (https://www.crd.york.ac.uk/prospero/#myprospero).

### Search strategy and information source

The adapted PECO format was used to explicitly review the literature and clarify the specifications of the inclusion and exclusion criteria. The adapted PICO comprises population (P), exposure (E), outcome (O), and context (setting as described below).

a. **Population**: children

b. **Exposure**: Associated factors, predictors, risk factors

c. **Context (Setting)**: Countries in Africa

d. **Outcome**: Childhood traditional uvulectomy (TCU)

   Therefore, by using this adapted PICO format, the following review questions were formulated for the search for primary studies.

1. What is the pooled prevalence of traditional childhood uvulectomy in Africa?

2. What are the associated factors with traditional childhood uvulectomy in Africa?

   Based on the aforementioned review PICO format and questions, primary studies performed from MEDLINE, Science Direct, Google Scholar, and African Journal Online were searched.

   The core search terms and phrases used were "prevalence", "incidence", "epidemiology", "proportion", "magnitude", "burden", "predictors", "risk factors", "associated factors" and "Africa". The search strategies were developed using different Boolean operators. Notably, to fit the advanced MEDLINE database, the following search strategy was applied on 6 November-23 December, 2023: [(prevalence[MeSH Terms]) OR incidence[MeSH Terms]) OR proportion [MeSH Terms]) OR epidemiology[MeSH Terms]) OR magnitude[MeSH Terms]) OR burden[MeSH Terms]) AND predictors [MeSH Terms]) OR risk factors [MeSH Terms]) OR associated factors [MeSH Terms]) AND (Africa)].

### Study selection

Primary studies downloaded and retrieved from the databases were exported to Mendeley Desktop 1.19.8 reference manager software to remove duplicate studies. The study selection process was held in two stages. First, the title and abstract were screened, and second, a full-text review was performed. Two independent reviewers (KA and DK) screened the title and abstract. Disagreements were resolved based on established article selection criteria, and discussion with the third author (SD) resolved any disagreements to decide whether the study was eligible for inclusion or not. Two independent authors (TMA and AK) reviewed the abstracts and full texts of eleven articles published between 2010 and 2024.

### Eligibility criteria

   **Inclusion criteria.** Included studies were articles that reported the prevalence of traditional childhood uvulectomy in general and/or at least one or more factors in Africa. Only studies published in English and in Africa were included.

**Exclusion criteria.** Articles without full text available, adult uvulectomy, or qualitative studies were excluded.

## Quality assessment

**Critical appraisal and quality assessment.** The risk of bias assessment for the included studies was conducted using the quality appraisal criteria from the Joanna Briggs Institute (JBI) [25]. The JBI checklist includes a series of quality indicators that assess the methodological rigor of each study. Studies were considered to be at low risk of bias if they scored 6 or above out of 9 on the quality assessment checklist. By using the JBI checklist, we ensured that all included studies underwent a consistent and reliable assessment of their methodological quality, which helps to enhance the validity and credibility of the findings. Two independent authors (SD and DK) appraised the quality of the studies. Disagreements were resolved by consulting a third reviewer (KA). The quality scores for the primary studies were reported (S2 File).

## Data extraction

The data were determined to be extracted based on two criteria: 1. Clear and consistent operational definitions for the dependent variable (traditional childhood uvulectomy) and 2. These variables are statistically associated with the outcome variable reported by the AOR (S3 File).

Two authors extracted the data using the standardized format of the MS Excel spread sheet. The name of the first author and year, country name, study setting, study design, sample size, sample category, prevalence, and odds of predictors were extracted. Whenever variations were observed, the phase was repeated. If discrepancies between the data extracted were present, a third reviewer was involved in the decision (SD). After reporting the inconsistency of the data in the primary study, data transformation was conducted.

## Outcome measurement

### Traditional childhood uvulectomy

Traditional Childhood Uvulectomy (TCU) is a cultural practice in which a portion of the uvula (the small, fleshy extension at the back of the soft palate) is surgically removed or altered by traditional healers or unqualified practitioners, typically without medical supervision on under 15 age group in Africa [2,7,26].

### Statistical analysis

The required data were collected using the Microsoft Excel 2013 workbook. Then, STATA version 17 statistical software was used for the meta-analysis. Publication bias was objectively checked using Egger's regression test. The heterogeneity of the studies was quantified using the I-squared statistic. Both pooled prevalence analysis and pooled effect of predictors were conducted using a weighted inverse variance random-effects model. This model was selected because it is better suited to handle the variability between studies, providing a more nuanced and reliable estimate of the pooled prevalence of TCU and its predictors across Africa, compared to a fixed-effects model. Subgroup analysis was conducted based on the following factors: country (Ethiopia, which had a higher concentration of primary studies, and other African countries), study setting (community-based or health facility-based), and sample size (categorized as less than 500 or 500 and above).

Missing data was handled by using random effect model and sensitivity analysis. A random-effects model minimizes the impact of missing data by accounting for study variations and prioritizing reliable data. Sensitivity analysis tests different assumptions, such as

imputing missing values or excluding incomplete studies, to ensure robust and transparent meta-analysis results.

### Ethics statement

Not applicable

## Results

### Selection of studies for review

The search strategy retrieved 46 articles, 16 of which were identified as duplicates and removed. Among the screened articles, 30 were from PubMed (n = 12), Google Scholar (n = 10), and the African Health Journal Repository (n = 8) (S4 File). The full texts of twenty-one articles were reviewed, and eleven studies were included in the review and meta-analysis (Fig 1).

### Characteristics of the included studies

A total of eleven cross-sectional studies with a pooled sample size of 7,231 children were included in this meta-analysis. Seven studies were found in each part of Ethiopia, particularly two from the Northwest Ethiopia between 2020–2022 [2,11], two from the North Ethiopia in 2013 [5,27], one from the Central Ethiopia in 2016 [8], two from the Southern Ethiopia between 2010–2023 [9,28], two from Nigeria in the Northwest and Plateau states between 2011–2016 [7,29] and one each from the Democratic Republic of the Congo and Niger between 2018–2023 [16,30] (Table 1).

The maximum sample size was 1163, and the minimum sample size was 402 among the reviewed studies, for a total of 7231 children [9,28]. Traditional childhood uvulectomy was not uniformly practiced across Africa; its prevalence varies significantly by region and community. This practice is more common in some places than in others according to this review. Regarding this unequal distribution, the study settings for articles from Ethiopia were as follows: two articles from the Northwest, particularly from Debre Tabor [2,11]; two from southern Hawassa and Gamo [9,28]; two from Northern Axum town by [27]; and one from central region in Debre Birhan town [8].

One study from another part of Africa was from the DRC in South Kivu [16], two studies from Nigeria in the Northwest region and plateau state [7,29], and one study from Niger in Niamey [30].

The studies included in this systematic review and meta-analysis had no considerable risk of bias. Therefore, all the studies were included. We assessed the studies with a JBI risk of bias quality appraisal checklist for systematic review [25]. According to this appraisal checklist, none of the included studies were of poor quality and all included studies had a quality score ≥6 out of 9 criteria. Tables and figures were used to visually display the findings. Ten studies were excluded from the review because they did not fit the outcome measures specified for this review or scored below 6 out of 9 on the quality assessment appraisal. Additionally, qualitative studies were excluded upfront as part of the exclusion criteria. The ten excluded studies were identified after full-text appraisal (Table 2).

## Meta-analysis

### The pooled prevalence of TCU

The pooled prevalence of TCU from eleven studies [2,5,7–9,30,11,16,27–29] was 40.98 (95% CI = 25.04–56.92; I² = 99.61, P = 0.001) (Fig 2).

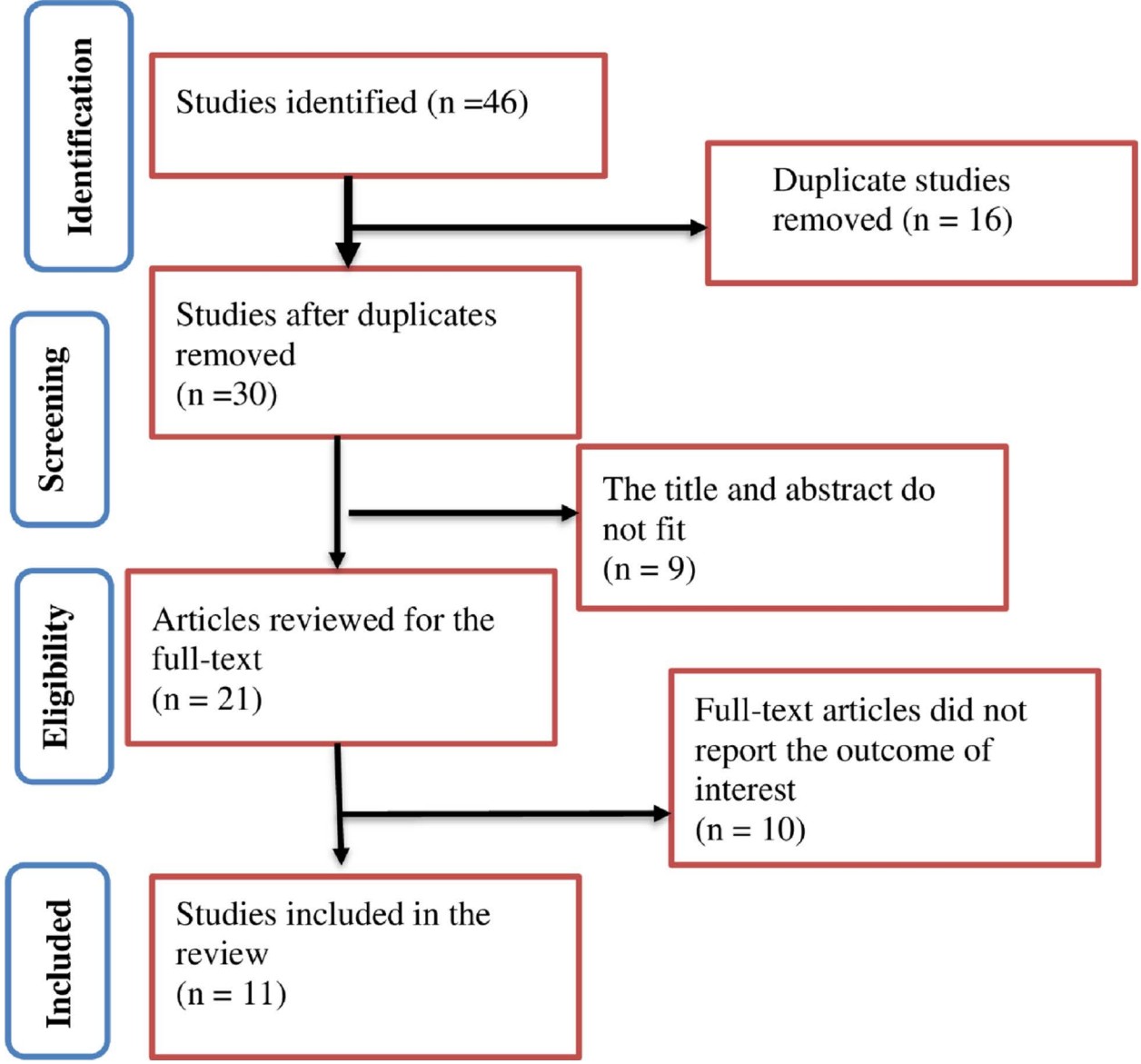

**Fig 1. PRISMA flow diagram for pooled prevalence and associated factors of TCU in Africa, 2023.**

The absence of publication bias was assessed using Egger's regression test. A significant result (P < 0.05) suggests potential publication bias, while a non-significant result (P > 0.05) indicates no evidence of bias. In this analysis, Egger's test yielded p = 0.098, indicating no significant evidence of publication bias (Table 3).

**Sensitivity analysis.** A sensitivity analysis was conducted to evaluate the stability and reliability of the findings in this systematic review and meta-analysis. This involved systematically excluding individual studies to assess their impact on the pooled effect size. The results showed that the effect sizes of all included studies ranged from 36.37 to 43.49 (95% CI), which fell within the overall pooled effect size range of 25.04 to 56.92 (95% CI). These findings indicate that no single study had a disproportionate influence on the overall results, confirming the robustness and consistency of the meta-analysis (Fig 3).

**Table 1. Characteristics of the studies reviewed on the prevalence and associated factors of TCU in Africa, 2023.**

| No | Author | Country | Study setting | Sample size category | Study design | Age group | Region category | Study area |
|---|---|---|---|---|---|---|---|---|
| 1 | Adoga., et al. (2011) | Nigeria | Health facility based | >500 | Cross-sectional | Under 10 | Nigeria | Jos |
| 2 | Bayih et al. (2020) | Ethiopia | Health facility based | <500 | Cross-sectional | Neonates | Northwest | Debre Tabor |
| 3 | Djakounda, et al. (1994) | Niger | Health facility based | >500 | Cross-sectional | Under 5 | Central | Niamey |
| 4 | Farouk et al. (2023) | DRC | Health facility based | >500 | Cross-sectional | Under 15 | South Kivu | South Kivu |
| 5 | Gebrekirstos et al. (2013) | Ethiopia | Community-based | >500 | Cross-sectional | Under 5 | Northern | Axum |
| 6 | Gebrekrstos et al. (2014) | Ethiopia | Community-based | >500 | Cross-sectional | Under 5 | Northern | Axum |
| 7 | Kebede et al. (2017) | Ethiopia | Community-based | <500 | Cross-sectional | Under 5 | Central | Debre Birhan |
| 8 | Kefelew E. (2023) | Ethiopia | Health facility based | <500 | Cross-sectional | Under 5 | Southern | Gamo |
| 9 | Mitke YB (2010) | Ethiopia | Community-based | >500 | Cross-sectional | Under 5 | Southern | Hawassa |
| 10 | Oluwatosin et al. (2016) | Nigeria | Health facility based | >=500 | Cross-sectional | Under 15 | Northwest | Nigeria |
| 11 | Yirdaw BW (2022) (3) | Ethiopia | Community-based | >500 | Cross-sectional | Under 5 | Northwest | South Gondar |

Data extracted by KA, AK, DK, and TMA from November 6 to December 23, 2023, with eligibility confirmation by SDK.

**Table 2. List of excluded studies from the review of pooled prevalence and associated factors of TCU in Africa, 2023.**

| Author, Year | Reason for Exclusion | Explanation |
|---|---|---|
| Abera B., 2024 | Did not report outcome variables | The study involved populations that did not meet the inclusion criteria of uvulectomy. |
| Hadush, A., 2016 | Did not report outcome variables | The study focused on neonatal care, maternal knowledge, and other complications, not uvulectomy. |
| Kunii, O., 2006 | Excluded by quality assessment appraisal | The study examined all traditional practices in a refugee camp, which did not align with the review objectives. |
| Ndu, I. K., 2022 | Out of the review scope | The study focused broadly on developing countries, not specifically on African settings. |
| Kiflu, G., 2014 | Excluded by quality assessment appraisal | The study covered tooth extraction, uvulectomy, and female genital mutilation, lacking specific focus on uvulectomy. |
| Addis, G., 2002 | Did not meet outcome variables | The study was methodologically inappropriate, with qualitative narration rather than measurable outcomes. |
| Dagnew, M. B., 1990 | Did not report outcome variables | The study broadly classified traditional practices on children without focusing specifically on uvulectomy. |
| Belachew, A., 2020 | Did not report outcome variables | The study examined infection transmission in various traditional procedures, including uvulectomy, without detailed focus. |
| Hailu, A., 2019 | Did not report outcome variables | The study centered on awareness of throat infections and complications, not uvulectomy. |
| Mamuye B., 2020 | Did not report outcome variables | The study was methodologically unsuitable for inclusion in the review and meta-analysis. |

Data extracted by KA, AK, DK, and TMA from November 6 to December 23, 2023, with eligibility confirmation by SDK.

## Subgroup analysis

Subgroup analysis was conducted based on the following factors: country, study setting, and sample size. For the country factor, studies were grouped into those conducted in Ethiopia, which had a higher concentration of primary studies, and other African countries, including Nigeria, the Democratic Republic of Congo (DRC), and Niger.

For the study setting, studies were categorized as community-based or health facility-based. Community-based studies focused on populations living in their residential areas, reflecting traditional practices and behaviors in their natural environments. In contrast, health facility-based studies examined populations accessing modern healthcare services, spanning primary to tertiary care levels.

Regarding sample size, studies were grouped into those with a sample size of less than 500 and those with 500 or more participants to assess the potential influence of study population size on the findings.

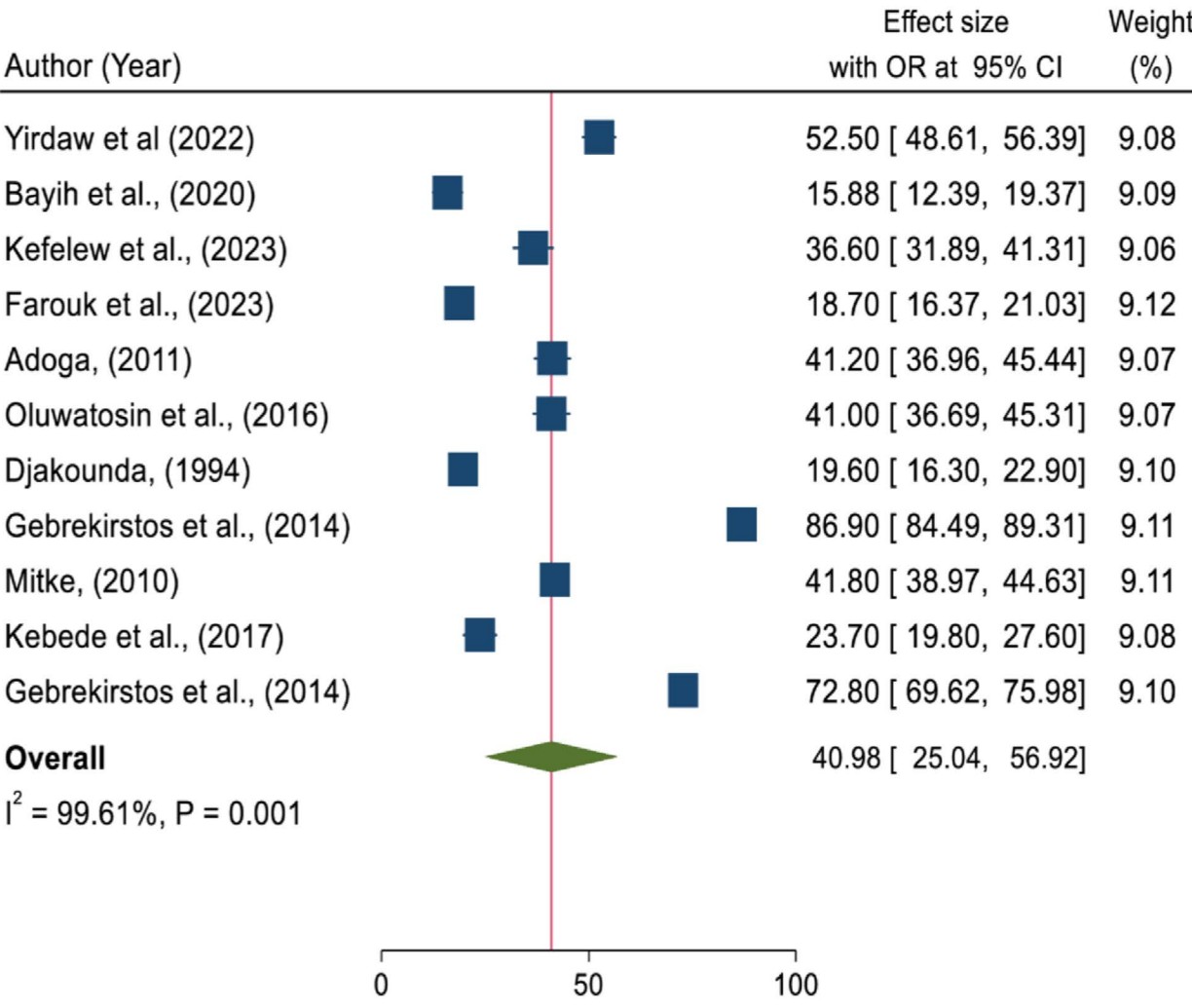

**Fig 2. Forest plot for pooled prevalence of TCU in some African countries, 2023.**

**Table 3. Egger's test for publication bias for prevalence and associated factors of TCU in Africa, 2023.**

| Std.-Eff. | Coef. | Std. Err | t | p>t | 95% confidence interval | |
|---|---|---|---|---|---|---|
| Slope | −12.06589 | 18.557902 | −0.65 | 0.532 | −54.04932 | 29.91754 |
| Bias | 62.57097 | 33.95122 | 1.84 | 0.098 | −.14.23202 | 139.374 |

However, there was no effect on reducing the heterogeneity of the heterogeneity ($I^2$). According to the seven studies of pooled effects in Ethiopia, the incidence of TCU was 47.19 (26.28–68.10), and according to four studies in other African countries, the incidence of TCU was 30.03 (18.11–41.94). After subgroup analysis by study setting, five studies used a community-based cross-sectional study design, for which the pooled prevalence was 55.56% (32.93–78.20). The remaining six studies used a health facility-based cross-sectional study design and had a pooled prevalence of 28.73% (19.67–37.78). The pooled prevalence of TCU in studies that used health facilities as a source of data was lower than that in studies obtaining data from the community. According to our subgroup analysis by sample size, the pooled

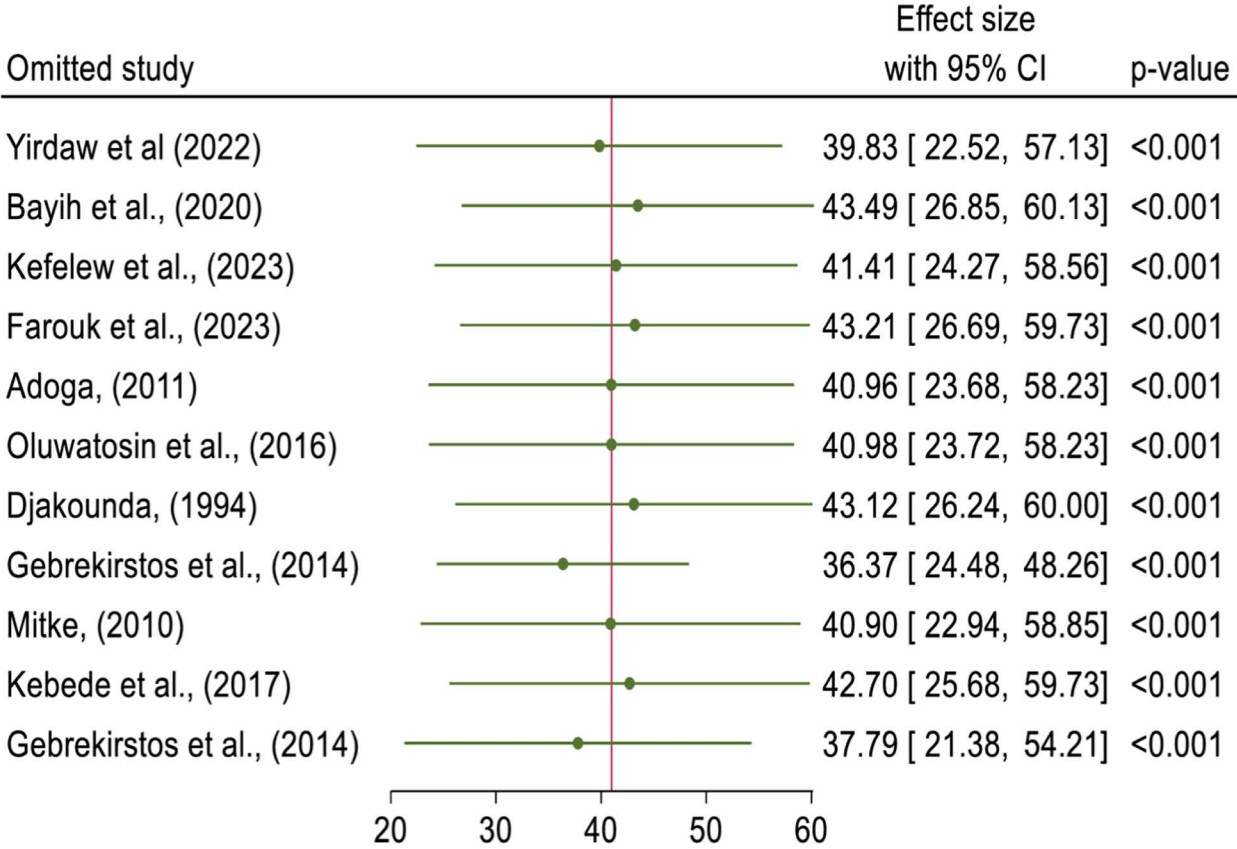

**Fig 3. Sensitivity analysis for pooled prevalence and associated factors of TCU in Africa, 2023.**

incidence in three studies with a sample size less than 500 was 25.30% (13.97–36.63). The pooled incidence was 47.65% (26.04–69.25) for studies with sample sizes greater than 500 in seven studies (Table 4).

## Factors associated with TCU

Based on this review, the predictors of traditional childhood uvulectomy according to pooled data from two studies were identified as rural residents and mothers with older children who had a previous history. By pooling studies, rural resident mothers were 2.45 times more likely to have a child who undergo traditional childhood uvulectomy (TCU) compared to mothers living in urban areas [8,9] (AOR = 2.45; 95% CI = 1.59–3.32). Mothers with a history of previous traditional childhood uvulectomy on their children were 8 times more likely to underwent TCU than were those with no history of uvulectomy (AOR = 8.44; 95% CI = 6.27–10.61) [2,11] (Table 5).

In a study by Yirdaw et al., mothers who lacked information about uvulectomy and those who perceived the uvula as a cause of infection were significantly more likely to have a child undergo traditional childhood uvulectomy (TCU) compared to mothers who were informed about its complications. Specifically, mothers lacking information were nearly three times more likely to have a child who underwent traditional childhood uvulectomy (TCU) (AOR = 2.98; 95% CI: 2.40–3.56), while those who believed the uvula caused infection were almost five times more likely to do so (AOR = 4.89; 95% CI: 4.64–5.14) [11]. According to maternal

**Table 4. Subgroup analysis for pooled prevalence and associated factors of TCU in Africa, 2023.**

| Sub-group | Author and year | Pooled OR (95% CI) | I² (P value) |
|---|---|---|---|
| By Ethiopia | (Gebrekirstos et al., 2014), (Mitke, 2010), (Kebede *et al.*, 2017), (Yirdaw et al., 2022), (Gebrekirstos et al., 2014), (Bayih et al., 2020) | 47.19 (26.28–68.10) | 99.64% (0.00) |
| By Other African countries (DRC, Niger and Nigeria) | (Farouk et al., 2023), (Kambale *et al.*, 2018), (Adoga, 2011b), (Oluwatosin *et al.*, 2016), (Djakounda, 1994) | 30.03 (18.11–4194) | 97.96% (0.00) |
| By community based (data collected from community) | (Kebede *et al.*, 2017), (Yirdaw et al., 2022), (Gebrekirstos et al., 2014), (Mitke, 2010) | 55.56 (32.93–78.20) | 99.62% (0.00) |
| By Health facilities (from primary health care to tertiary level) | (Farouk et al., 2023), (Kambale *et al.*, 2018), (Adoga, 2011b), (Oluwatosin *et al.*, 2016), (Djakounda, 1994), (Bayih et al., 2020), (Kefelew et al., 2023) | 28.73 (19.67–37.78) | 97.45% (0.00) |
| Sample size > 500 | (Bayih et al., 2020)(Kebede *et al.*, 2017), Kefelew et al., 2023) | 25.30 (13.97–36.63) | 95.84% (0.02) |
| Sample size < 500 | (Farouk et al., 2023), (Kambale *et al.*, 2018), (Adoga, 2011b), (Djakounda, 1994), (Gebrekirstos et al., 2014), (Mitke, 2010) (Yirdaw et al., 2022), (Gebrekirstos et al., 2014) | 47.65 (26.04–69.25) | 99.72% (0.00) |

Data extracted by KA, AK, DK, and TMA from November 6 to December 23, 2023, with eligibility confirmation by SDK. Missing data were handled with a random effects model and sensitivity analysis.

**Table 5. Factors associated with TCU by maternal residency and history of having a child with previous TCU in Africa, 2023.**

| Factors associated with TCU | Pooled OR at 95% CI | I² (%) | P value |
|---|---|---|---|
| Maternal rural residency | 2.45 (1.59–3.32) | 0.01 | 0.042 |
| Having a child with previous TCU experience | 8.44 (6.27–10.61) | 81.71 | 0.002 |

Data extracted by KA, AK, DK, and TMA from November 6 to December 23, 2023, with eligibility confirmation by SDK.

attitudes, children with uvulitis (inflammation and swelling of the uvula) that did not respond to medical treatment were nearly four times more likely to undergo traditional uvulectomy compared to their counterparts (AOR = 3.92; 95% CI: 3.28–4.56) [11].

According to other studies by Bayih et al., children who were home-delivered and male neonates were 6 and 4.56 times more likely to receive TCU, respectively, than were their counterparts (AOR = 6.02; 95% CI = 4.36–7.68) and at (AOR = 4.56; 95% CI = 3.08–6.06), respectively [2]. According to studies by Kebede et al. and Kefelew et al., housewife mothers and those with unfavourable attitudes toward modern medical care were nearly three and four times more likely to have a child with TCU, respectively, than were their counterparts and positive attitudes toward health care (AOR = 2.98; 95% CI = 1.72–4.20) and AOR = 4.31; 95% CI = 2.17–6 [8,9] respectively.

## Discussion

The pooled prevalence of TCU was notably high at 40.98% (95% CI: 25.04%–56.92%) in certain African countries, with significant variability observed both between countries and across regions within the same country. This finding is comparable, though slightly lower, than the 44% pooled prevalence reported by Getachew et al. (2024), with a 95% CI of 31%–57%[3]. Although this review did not describe the specific traditional medicines used, the most common users of traditional medicine were identified in India, China, and Africa. This prevalence may be attributed to the widespread reliance on traditional medicine by the majority of the population in these regions. In Africa, traditional medicine has historically been the dominant healthcare system, deeply rooted in cultural practices long before the introduction of modern medicine [18,31].

This meta-analysis revealed that rural residents were three times more likely to practice traditional childhood uvulectomy (TCU) compared to their urban counterparts. This disparity can be attributed to several factors. In rural areas, traditional beliefs and practices are often more deeply entrenched, with communities relying on local customs and traditional healers for healthcare [8,9]. This might be because socioeconomic status plays a role in choosing traditional medicine in the community over modern health care [32]. From a modern health-seeking perspective, rural residents might have reduced willingness and poor infrastructure to access health care compared to urban residents, so they prefer to use traditional healers found nearby in their village [4,17,33,34]. Health-seeking behaviors in rural communities might also hinder parents' choice of care during childhood illness. Traditional medicine is deeply embedded in cultural identity, with parents often favoring herbal treatments or spiritual interventions over conventional medical care. This reliance on traditional practices can delay the timely treatment of childhood illnesses, leading to complications and worsening health outcomes. Additionally, there may be a lack of awareness or mistrust of modern medical practices, further reinforcing the preference for traditional remedies [1,15,32,35].

Having a previous history of traditional childhood uvulectomy (TCU) was associated with an eightfold increase in the likelihood of practicing TCU compared to individuals without such a history [2,11]. This finding may help explain why experienced users of traditional medicine often become advocates within their communities, presenting evidence of the 'safety' of traditional childhood uvulectomy (TCU) to new users. In many developing nations, traditional medicine is widely accepted for several reasons: the perceived ineffectiveness of modern medical treatments, beliefs regarding the incurability of certain diseases through contemporary healthcare, and the confidence in the efficacy of traditional remedies. Additionally, traditional medicine is often more affordable and culturally resonant, leading to a preference for these practices. Many individuals also feel embarrassed or stigmatized when seeking care from modern healthcare facilities, further reinforcing their reliance on traditional healing methods [10,15,36].

## Strengths and limitations

This review provides key estimates and predictors of Traditional Childhood Uvulectomy (TCU) in Africa, offering evidence-based insights to guide policymakers and healthcare professionals in addressing TCU-related health issues. These findings can help prioritize strategies to reduce under-5 morbidity and mortality in maternal and child health (MCH) settings. A major strength is the larger sample size compared to previous studies, providing a better understanding of TCU's burden in Africa. However, the review is limited by the scarcity of research on TCU's true burden and the heterogeneity of data sources, which may affect the generalizability of the findings.

## Conclusion

Traditional childhood uvulectomy (TCU) was prevalent in Africa, with a pooled prevalence of nearly 41% from eleven studies, higher than in other regions. Mothers in rural areas and those who have a child with a history of TCU in a previous pregnancy were more likely to engage in the practice. Therefore, it might be essential to examine its impact on childhood health admissions to fully understand the extent of the problem. Based on these findings, a targeted education outreach program should be developed within maternal and child health (MCH) initiatives to prevent TCU. This program could focus on raising awareness about the health risks associated with TCU and promote alternative practices. By integrating TCU prevention into MCH strategies, the project can help reduce the prevalence of TCU, mitigate its negative health outcomes, and ultimately improve maternal and child health.

## Supporting information

**S1 File 1. PRISMA checklist.**
(DOCX)

**S2 File 2. Quality assessment by JBI quality appraisal criteria.**
(DOCX)

**S3 File 3. Raw dataset for the manuscript.**
(XLSX)

**S4 File 4. Studies identified in literature search.**
(DOCX)

## Author contributions

**Conceptualization:** Solomon Demis Kebede.

**Data curation:** Kindu Agmas, Demewoz Kefale, Amare Kassaw, Tigabu Munye Aytenew.

**Formal analysis:** Solomon Demis Kebede, Demewoz Kefale, Amare Kassaw.

**Investigation:** Kindu Agmas, Tigabu Munye Aytenew.

**Methodology:** Solomon Demis Kebede, Demewoz Kefale.

**Software:** Tigabu Munye Aytenew.

**Supervision:** Kindu Agmas, Amare Kassaw.

**Validation:** Tigabu Munye Aytenew.

**Writing – original draft:** Solomon Demis Kebede.

**Writing – review & editing:** Solomon Demis Kebede.

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
