## [Decision Letter · Decision Letter 0]

2 Oct 2024

PONE-D-24-36948Prevalence of traditional uvulectomy and associated predictors among children in Africa: A systematic review and meta-analysisPLOS ONE

Dear Dr. Demis,

Thank you for submitting your manuscript to PLOS ONE. After careful consideration, we feel that it has merit but does not fully meet PLOS ONE’s publication criteria as it currently stands. Therefore, we invite you to submit a revised version of the manuscript that addresses the points raised during the review process.

We look forward to receiving your revised manuscript.

Kind regards,

Kahsu Gebrekidan, Ph.D.

Academic Editor

PLOS ONE

The American Journal Experts (AJE) (https://www.aje.com/ ) is one such service that has extensive experience helping authors meet PLOS guidelines and can provide language editing, translation, manuscript formatting, and figure formatting to ensure your manuscript meets our submission guidelines. Please note that having the manuscript copyedited by AJE or any other editing services does not guarantee selection for peer review or acceptance for publication.

3. In this instance it seems there may be acceptable restrictions in place that prevent the public sharing of your minimal data. However, in line with our goal of ensuring long-term data availability to all interested researchers, PLOS’ Data Policy states that authors cannot be the sole named individuals responsible for ensuring data access (http://journals.plos.org/plosone/s/data-availability#loc-acceptable-data-sharing-methods ).

5. We note that there is identifying data in the Supporting Information file < Supplementary file 2.pdf>. Due to the inclusion of these potentially identifying data, we have removed this file from your file inventory. Prior to sharing human research participant data, authors should consult with an ethics committee to ensure data are shared in accordance with participant consent and all applicable local laws.

-Location data

Additional guidance on preparing raw data for publication can be found in our Data Policy (https://journals.plos.org/plosone/s/data-availability#loc-human-research-participant-data-and-other-sensitive-data ) and in the following article: http://www.bmj.com/content/340/bmj.c181.long .

Reviewers' comments:

Reviewer's Responses to Questions

**Comments to the Author**

1. Is the manuscript technically sound, and do the data support the conclusions?

Reviewer #1: Partly

Reviewer #2: Yes

Reviewer #3: Partly

2. Has the statistical analysis been performed appropriately and rigorously?

Reviewer #1: Yes

Reviewer #2: Yes

Reviewer #3: I Don't Know

3. Have the authors made all data underlying the findings in their manuscript fully available?

Reviewer #1: Yes

Reviewer #2: Yes

Reviewer #3: Yes

4. Is the manuscript presented in an intelligible fashion and written in standard English?

Reviewer #1: No

Reviewer #2: Yes

Reviewer #3: No

5. Review Comments to the Author

Reviewer #1: A. Summary

This paper describes a systematic review and meta-analysis of the Prevalence of traditional childhood uvulectomy (TCU) and associated predictors in Africa. The literature search included publications until December 2023. The topic is important and relevant, and the findings are potentially useful to delineate the extent of the TCU and inform programmes which seek to address it. The paper also included a subgroup analyses, grouping Ethiopia separately to other African countries, however the rationale was not stated.

Although there is potential value of this paper, the actual value and interpretation of this paper is limited by the absence or lack of detail of multiple aspects which should be included as standard, according to PRISMA guidelines. In addition, the identification of less than half the number of studies in Ethiopia compared to the 2023 systematic review of Ethiopian studies by Getachew et al. (International Journal of Pediatric Otorhinolaryngology 176 (2024) 111835), is concerning, as well as the lack of reference to this study, since it was published online in December 2023.

The detailed review which follows, does not refer to line numbers, since none were provided, despite this being a requirement by PLOS ONE. If all aspects in the detailed review are adequately addressed, and if the subheadings used by the PRISMA guidelines are used in the abstract and the full manuscript the paper may be considered for publication.

B. Detailed review

1. Abstract

1.1 The abstract is missing multiple aspects which are required by PRISMA guidelines – the use of the subheadings listed in the PRISMA abstract check list is recommended, to ensure all aspects are included.

1.2 The following aspects must be addressed:

1.2.1 The standard adhered to (PRISMA) should be stated.

1.2.2 The objectives need to be explicitly stated, describing the outcome (TCU)

1.2.3 The inclusion criteria need to be explicitly stated

1.2.4 The methods need to be briefly described.

1.2.5 The information sources/databases need to be stated

1.2.6 The risk of bias methods need to be stated

1.2.7 The characteristics of studies need to be stated

1.2.8 The limitations of the evidence included need to be stated

2. Introduction/Background

2.1 The statement, “..in some parts of the world, including Africa, the majority of people depend on traditional medicine…”, should be more specific.

2.2 The last three paragraphs need to be referenced

2.3 The line, “A systematic intervention strategy in the current health care delivery system in Africa using the pooled prevalence and predictors of TCU could be launched”, needs to be rephrased as an aim and specific objectives must be stated, using the word, “objectives”

3. Methods

3.1 Add or clarify the following aspects:

3.1.1 How were studies grouped and why.

3.1.2 What were the dates of inclusion in the search (start date not mentioned), and what date was the search done.

3.1.3 The inclusion and exclusion criteria should be moved to after the review questions.

3.2 Search strategy and quality assessment

3.2.1 The limitation/Filter including studies only published in English and in Africa, should be stated as part of the search strategy.

3.2.2 The initials of the two independent authors who reviewed the abstracts and full texts of the eleven articles should be stated.

3.2.3 The search strategy only provided for Medline and is not complete, since no terms for uvulectomy were described – the full search strategy for all databases is required.

3.2.4 Studies were considered low risk when they fit quality assessment checklists of 6 or above – please expand and clarify this statement and provide more detail describing how risk of bias was assessed.

3.3 Outcome measurement and statistical analysis

3.3.1 Under outcome measurement, the statement, “all the procedures of cutting off a part of the uvula by traditional healers in the childhood age group (birth to under 15) in all African countries” appears to be missing a section – please edit and clarify.

3.3.2 What was the rationale for using the weighted inverse variance random-effects model?

3.3.3 Does “considering Ethiopia and other African countries” mean “comparing Ethiopia to other African countries”? Please clarify.

3.3.4 What categories were used for synthesis and why? (see also 3.1.1)

3.3.5 How were missing summary statistics handled ?

3.3.6 What methods were used to tabulate or visually display results of individual studies and syntheses?

3.3.7 What methods were used to assess risk of bias due to missing results in a synthesis (arising from reporting biases) ?

3.3.8 What p-values were used to interpret the publication bias assessment.

4. Results

4.1 Figure 1

4.1.1 The text in box, “The title and abstract do not fit” should be reworded more scientifically (eg “Title and/or abstract not relevant”)

4.1.2 The numbers of records per data base should be stated in the figure

4.2 Characteristics of included studies

4.2.1 Only seven studies in Ethiopia were identified, compared to 19 in the review by Getachew et al. (International Journal of Pediatric Otorhinolaryngology 176 (2024) 111835), despite the same eligibility criteria – please explain the discrepancy to the reviewer and ensure that this is discussed in the discussion.

4.2.2 The statement, “Traditional childhood uvulectomy was not practiced across Africa” is ambiguous. Does “across” mean “all” countries in Africa?

4.2.3 The table listing the characteristics of included studies should be referenced at the beginning of the paragraph.

4.2.4 The full term, “risk of bias” should be used in preference to “risk”.

4.2.5 The term “included” is preferable to “considered”, if indeed all were included.

4.2.6 The statement, “none of the included studies were of poor quality and had a quality score ≥6 out of 9 criteria” is not clear – did all have a quality score ≥6?

4.2.7 The reference to JBI quality appraisal checklist, should include the term, “risk of bias”.

4.2.8 The 9 quality criteria should be stated, and the scores should be included in table 1, or preferably separately in a graphic. (refer supplementary file 2 should be included as a table, with yes no low risk presented as graphics, rather than text.

4.2.8 Please add a table with details of the ten excluded studies which, “did not report the outcome of interest”.

4.3 Meta-analysis

4.3.1 It is more intuitive to the reader to first read about the pooled incidence of TCU, subgroup analysis and predictors , before describing the Egger’s test, so it is clear what outcomes it refers to.

4.3.2 Egger’s test assesses publication bias, not “absence of publication bias” – the text should briefly explain how the parameters in table show no publication bias.

4.3.3 The title of table 2 could rather be stated as, “Egger’s test for publication bias in the effect sizes of prevalence and predictors of TCU in Africa.”

4.3.4 What is meant by, “Mothers who were perceived as uvula causes”?

4.4 Subgroup analysis

4.4.1 More detail on which other African countries, types of communities and types of health facilities is needed.

4.4.2 The interpretation of the results with statements such as, “This implies that nearly half of the children would not seek medical care after they had received TCU”, should be moved from the results section to the discussion.

4.4.3 Table 3 needs a more detailed title explaining what it represents. Needs footnotes with abbreviations and explanations of the subgroups – eg, which other African countries, what does community-based mean? What does health facilities mean?

4.4.4 The associated forest plots for the subgroup analyses should be included in the text.

4.5 Predictors of TCU

4.5.1 Please correct the grammar of the statement, “nearly fourfold more patients with uvulitis not cured by medical treatment not cured by medical treatment than did their counterparts”

4.5.2 Please correct the grammar/syntax of the statement, in the last line of the results, “AOR=4.31; 95% CI=2.17-6[16,18]/>, respectively).”

5. Discussion

5. 1 The discussion should begin with statement referring to what was described in the study, rather than repeating the aim.

5.2 Please discuss the results for the Ethiopian studies in comparison to the systematic review of TCU in Ethiopia by Getachew et al.

5.3 The predictors of TCU were rural residency and a history of previous practice – why was maternal education not included, since this was identified in the systematic review of TCU in Ethiopia by Getachew et al.?

5.4 The statement “the most common utilizers of traditional medicine were found in India, China, and Africa”, appears to refer to the current study, which was limited to Africa – there is a need for more clarity when you refer to your study vs. others.

5.5 What were the strengths and limitations of your study?

5.6 How can the findings of your study be used to develop a project addressing TCU?

6. Conclusion

6.1 The word “common” is preferable to “quite common”.

6.2 Forty-one% should be written as 41%.

6.3 The last two sentences of the conclusion require grammatical correction, and some elaboration. 6.4 The conclusion in the text should be aligned with the conclusion in the abstract.

Reviewer #2: Dear Authors,

Thank you for submitting your manuscript. I have reviewed it and provided comments to enhance its quality and clarity. Your work makes an important contribution to understanding traditional childhood uvulectomy in this area, a topic that is significant in both cultural and health contexts. I recommend addressing all the comments provided to strengthen the manuscript and increase its impact.

Yours sincerely,

Reviewer #3: Abstract

Mothers who practice – perhaps use another word – “seek” or “request” = Currently it reads the mothers themselves do the cutting.

Outcome measures: TCU is all the procedures for cutting a part of the uvula by traditional – rephrase, (Example TCU includes all procedures for removing all or part of the uvula)

Results: pooled estimates of TCU in Africa was 40.98% - is this so for all African countries. This statement may be overinflating the true estimate of TCU prevalence. I am not familiar with this practice occurring in southern Africa.

Mothers with a history of previous TCU practice were 8.44 times more likely to practice than – rephrase for clarity (Example, mothers who requested TCU for a previous child were more likely to request TCU for a second child…) Is it only the mothers requesting TCU be performed, or anyone in the family?

Fortunately, all uvulectomy practices did not cause hospitalization – rephrase (Example, Not all children required hospitalization post TCU…)

Therefore, the problem should be prioritized – rephrase (Example, This practice must be …

Background

Traditional childhood uuvlectomy is an unsupported, illogical, and possibly a dangerous cultural malpractice procedure for all childhood sickness, including feeding, breathing, and related diseases. – rephrase for clarity – avoid stigma associated with words like illogical, malpractice…

Moreover, other common complications of traditional children uvulectomy consists of anaemia – rephrase – uvulectomy in children…

Thus, childhood admissions – rephrase – paediatric hospital admissions…require treatment such as…

In the African context, the community mistakenly attributed nearly all childhood illnesses to uvula so that ill children are often subjected to traditional uvulectomy because of parental perception of poor outcomes in modern medicine. Rephrase for clarity (Example. Traditional beliefs linking childhood illnesses to the uvula and parental lack of trust in modern medicine….

As per studies on childhood traditional uvulectomy in several African countries individually; it affects several children through the cutting of a part of the uvula by traditional healers. – rephrase.

Despite its significant clinical practical benefits, the pooled prevalence of TCU practice and predictors for TCU in Africa were not appropriately incorporated in the prevention of neonatal and under5 mortality. – rephrase. Here it reads TCU has a clinical benefit. Correct under five mortality…

A systematic intervention strategy in the current health care delivery system in Africa using the pooled prevalence and predictors of TCU could be launched – move sentence to recommendations or discussions section

Methods

Whenever variations are observed, the phase is repeated – past tense: were, was

all the procedures of cut… Capitalize first letter.

Included studies

Seven studies were found in each part of Ethiopia, particularly two from the Northwest[1,3], two from the North[14,15], one from the Central Hemisphere[16], two from the Southern Hemisphere[17,18], two from Nigeria in the Northwest and Plateau states[19,20] and one each from the Democratic Republic of the Congo and Niger[8,21]. – This sentence indicates prevalence in four African countries – not the whole of the continent. Also, Ethiopia is not in the southern hemisphere – consider replacing the word “hemisphere”

Traditional childhood uvulectomy was not practiced across Africa. This practice is more common in some places than in others according to this review. – move to discussion section

…had no considerable risk. Specify risk of bias or other risk.

Subgroup analysis

However, there was no effect on reducing the heterogeneity of the heterogeneity (I2 ) – rephrase for clarity…. Heterogeneity of the heterogeneity?

This implies that nearly half of the children would not seek medical care after they had received TCU – consider replacing seek with require

Predictors

Based on this review, the predictors of traditional childhood uvulectomy according to pooled data from two studies were identified as rural residents and had a previous history of traditional rephrase – had previous history of TCU reads as the same child had a second procedure done.

Mothers with a history of previous traditional childhood uvulectomy were 8 times more likely to practice than were those with no history of uvulectomy – did mothers have the uvulectomy, or they have a previous child who underwent TCU? Do mother practice TCU or seek TCU for their child?

In a single study by Yirdaw et al., mothers with a lack of information and who were perceived as uvula causes were almost – rephrase, incomplete thought/structure.

respectively, than were those informed about complications caused by TCU (AOR=2.98; 95% CI=2.40-3.56) and those informed about complications caused by TCU (AOR=4.89; 95% CI=4.64-5.14), respectively[1]. – remove duplication in the sentence.

According to maternal attitudes, nearly fourfold more patients with uvulitis (inflammation and swelling of the uvula) not cured by medical treatment than did their counterparts (AOR=3.92; 955CI=3.28-4.56)[1] – incomplete sentence

…children who were home-delivered and male neonates were 6 and 4.56 times more likely to practice TCU… rephrase (Example, receive TCU instead of practice)

…housewife mothers and those with unfavourable attitudes toward modern medical care were nearly three and four times more likely to practice TCU, respectively, than were their counterparts and positive attitudes toward health…- replace practice with seek or request, replace and using the word “with”.

Discussion

This might be because socioeconomic status plays a role in choosing traditional medicine in the community over modern health care[24]. - This sentence seems to contradict earlier statement which refers to traditional beliefs as distrust in the modern health care being reasons for undergoing TCU. Not accessibility issues. Socioeconomic status was not assessed and it may be questionable why this is referred to here as a possible cause.

Conclusion

Traditional childhood uvulectomy is quite common in Africa. – Rephrase – there may be a high prevalence in some African countries, but not all.

Thus, studies focused on the burden of the problem of childhood admissions caused by TCU are recommended to be more beneficial. Moreover, it was advised to develop a project involving the problem in maternal-child health (MCH) packages – expand these two thoughts to improve clarity.

References

6. Ijaz N. Pr ep rin t n ot pe er r ev iew Pr ep rin t n ot pe er ed. : 1–37. – check spacing

8. R MK, Balibuno Y, N IF, J BK, G FM, Masumbuko B. Uvulectomie traditionnelle , une pratique courante au Sud-Kivu en République démocratique du Congo Traditional uvulectomy , a common practice in South Kivu in the Democratic Republic of Congo – who translated the French into English for data extraction?

23. ajol-file-journals_209_articles_67959_submission_proof_67959-2485-140140-1-10- 20110715.pdf. – improve citation

27. ajol-file-journals_336_articles_58296_submission_proof_58296-4009-103363-1-10- 20100823.pdf – improve citation

General comments

The text may be improved by professional editing and grammar checking to strengthen the clarity the authors require.

TCU is not practiced equally across Africa, the authors need to clarify where this practice is prevalent and where it is not. Alternatively, the authors may need to restrict the analysis to the four listed countries.

It is unclear who did the translation for the French article included in the references.

The text does not report on strengths and limitations of the study, methodology or analysis.

Include the forest plots into the main text and reference them in the text.

Confusion arises when the text refers to mothers practicing TCU - rather than seeking or requesting TCU. Similarly, confusion arises when authors refer to mothers with previous history. It is unclear if previous history refers to maternal or another sibling's uvulectomy.

The text dominates focus on the mother - is she the only care provider seeking TCU or is this a family practice?

The discussion section can be strengthened by a short referral to why TCU remains prevalent, when other practices like circumcision are being addressed by WHO and health ministries. Additionally, to highlight the health risk and danger of TCU, it may be of interest to the reader to know the percentage of procedures resulting in mild, moderate and severe complications.

The reference section needs major review and standard citation practices need to be adhered to.

Thank you.

6. PLOS authors have the option to publish the peer review history of their article (what does this mean? ). If published, this will include your full peer review and any attached files.

**Do you want your identity to be public for this peer review?** For information about this choice, including consent withdrawal, please see our Privacy Policy .

Reviewer #1: No

Reviewer #2: **Yes: ** Seid Muhumed Abdilaahi

Departments of Pediatrics and Child Health Nursing, Jigjiga University, Jigjiga, Ethiopia

Reviewer #3: **Yes: ** Beatrix Callard

---

## [Author Response · Author response to Decision Letter 0]

2 Nov 2024

Authors’ response to reviewers:

Comment A1: Title Consistency: The title suggests a broader scope ("Africa") while the data focuses exclusively on Sub-Saharan Africa. This inconsistency needs to be addressed.

Authors’ response: Revised

Comment A5: This is not clear regarding which problem should be prioritized; further elucidation is needed.

Authors’ response: Traditional uvulectomy practices should be prevented, so priority in MCH services in sub Saharan Africa.

Comment A6: Revised

Comment A7: revised

Comment A8: Correct and concise. However, consider adding more detail about the uvula’s role in modern medicine (e.g., speech, immune defense) to frame the risks associated with removing it.

Authors’ response: revised

Comment A9: revised

Comment A10: revised

Comment A11: revised

Comments A12: revised

Comments A13: The phrasing here is quite strong and may come off as culturally insensitive. Consider revising to a more neutral tone and cite it.

Authors’ response: revised

Comment A 14: revised

Comment A15: revised

Comment A16: Listing complications is useful, nonetheless it would be more impactful if you provided specific prevalence rates or real-world examples from studies. How frequent are these complications in different regions?

Authors’ response: revised

Comment A17: revised

Comment A 18: revised

Comment A 19: This is a critical point that could benefit from more exploration. Are there specific studies linking traditional uvulectomy prevalence with socioeconomic factors? You could also discuss how access to formal healthcare influences the decision to use traditional medicine

Authors’ response: no, but low literacy arte and low utilization of health care is linked. It has been revised.

Comment A20: This is an interesting cultural observation, but it would benefit from citations or examples of studies or ethnographic accounts that explore these beliefs. Providing context for why this belief persists (e.g., mistrust of modern healthcare) would be helpful.

Authors’ response: revised

Comment A 21: revised

Comment A 22: This sentence feels repetitive given earlier statements. You could instead expand on why this practice persists in certain regions and explore whether any interventions have been attempted to curb it.

Authors response: revised

Comment A 23: revised

Comment A24: This sentence is unclear and somewhat contradictory. If the practice has significant risks, it wouldn't have clinical benefits. Consider rephrasing.

Authors’ response: revised

Comment A25: The exclusion criteria are well-stated. Would You justify why qualitative studies were excluded?

Authors’ response: Because it is empirically (quantitative) analysis of on its burden and associated factors. No studies using qualitative model were not considered.

Comment A26: It would be useful to explain what the scale of 6 or above means for readers unfamiliar with JBI criteria. What does a score of 6 indicate about the study’s reliability?

Authors’ response: revised

Comment A27: disagreements were some variation in selecting for articles, and appraising process. Truly, speaking there were no documented disagreement by third reviewer. But, he was modulating the process thoroughly.

Comment A28: The statement is clear, however you should specify the range of publication years for these studies. For example, mention whether these studies were published within a specific timeframe (e.g., between 2010 and 2024), which can provide additional context regarding the relevance and timeliness of the data.

Authors’ response: revised

Comment A29: revised

Comment A30: Why is there a specific focus on certain regions of Ethiopia? Is there any rationale behind the higher number of studies from this country compared to other African nations? It might be worth briefly discussing this in the results.

Authors’ response: this will be study objective in future, why some regions are more prevalent. But, in this review, we simply want to show prevalence and associated factors only.

Comment A31: This statement seems vague. Consider specifying which regions or countries practice traditional childhood uvulectomy the most and why there are regional differences. Is it tied to socioeconomic factors, education, or cultural beliefs?

Authors’ response: tried to response in Comment A30.

Comment A32: revised

Comment A33: The Egger’s test results indicate no publication bias, but further interpretation of these findings in relation to the overall analysis could enhance clarity.

Authors’ response: no publication bias but significant heterogeneity was observed and subgroup analysis was undergone for that case.

Comment A34: revised

Comment A35: Consider providing examples of specific traditional medicines used in different African cultures to add depth to this point.

Authors’ response: it is noteworthy that the most frequent users of traditional medicine are found in India, China, and various African nations. This widespread utilization can be attributed to traditional healing practices, such as herbal remedies, acupuncture, and spiritual therapies, which are integral to the healthcare systems in these regions. In many African countries, traditional medicine remains a dominant component of healthcare delivery, deeply rooted in cultural practices and beliefs that have persisted for generations, long before the introduction of modern medical approaches. So, there were no specific literatures for traditional uvulectomy for discussion purpose.

Comment A36: It may be useful to include more data or references that highlight the connection between socioeconomic status and traditional medicine use.

Authors’ response: revised, the available literatures are limited.

Comment A37: revised

Comment A38: revised

Comment A39: revised

Comment A40: What does this mean? Is it high or low, considering it is nearly 41%

Authors’ response: it is higher compared to other continents traditional practice.

Comments to the Author

5. Review Comments to the Author

Reviewer #1: A. Summary

This paper describes a systematic review and meta-analysis of the Prevalence of traditional childhood uvulectomy (TCU) and associated predictors in Africa. The literature search included publications until December 2023. The topic is important and relevant, and the findings are potentially useful to delineate the extent of the TCU and inform programmes which seek to address it. The paper also included a subgroup analyses, grouping Ethiopia separately to other African countries, however the rationale was not stated.

Although there is potential value of this paper, the actual value and interpretation of this paper is limited by the absence or lack of detail of multiple aspects which should be included as standard, according to PRISMA guidelines. In addition, the identification of less than half the number of studies in Ethiopia compared to the 2023 systematic review of Ethiopian studies by Getachew et al. (International Journal of Pediatric Otorhinolaryngology 176 (2024) 111835), is concerning, as well as the lack of reference to this study, since it was published online in December 2023.

The detailed review which follows, does not refer to line numbers, since none were provided, despite this being a requirement by PLOS ONE. If all aspects in the detailed review are adequately addressed, and if the subheadings used by the PRISMA guidelines are used in the abstract and the full manuscript the paper may be considered for publication.

Authors’ response: as studies in Ethiopia are much more compared to other countries and to have comparable sample size, we determine subgrouping by Ethiopia and other countries. May be appraisal tool variation and population variation based on objectives in the perspective studies.

B. Detailed review

1. Abstract

1.1 The abstract is missing multiple aspects which are required by PRISMA guidelines – the use of the subheadings listed in the PRISMA abstract check list is recommended, to ensure all aspects are included.

1.2 The following aspects must be addressed:

1.2.1 The standard adhered to (PRISMA) should be stated.

1.2.2 The objectives need to be explicitly stated, describing the outcome (TCU)

1.2.3 The inclusion criteria need to be explicitly stated

1.2.4 The methods need to be briefly described.

1.2.5 The information sources/databases need to be stated

1.2.6 The risk of bias methods need to be stated

1.2.7 The characteristics of studies need to be stated

1.2.8 The limitations of the evidence included need to be stated

Authors’ response: the recommended components of abstract were included, Limitation was included in the main body of the manuscript.

2. Introduction/Background

2.1 The statement, “..in some parts of the world, including Africa, the majority of people depend on traditional medicine…”, should be more specific.

2.2 The last three paragraphs need to be referenced

2.3 The line, “A systematic intervention strategy in the current health care delivery system in Africa using the pooled prevalence and predictors of TCU could be launched”, needs to be rephrased as an aim and specific objectives must be stated, using the word, “objectives”

Authors’ response: revised

3. Methods

3.1 Add or clarify the following aspects:

3.1.1 How were studies grouped and why.

Authors’ response: articles were grouped as Ethiopian studies and other African countries.

3.1.2 What were the dates of inclusion in the search (start date not mentioned), and what date was the search done.

Authors’ response: 6 November-23 December, 2023.

3.1.3 The inclusion and exclusion criteria should be moved to after the review questions.

Authors’ response: yes, it is after the review questions.

3.2 Search strategy and quality assessment

3.2.1 The limitation/Filter including studies only published in English and in Africa, should be stated as part of the search strategy.

Authors’ response: presented as recommended in inclusion criteria and in limitation section.

3.2.2 The initials of the two independent authors who reviewed the abstracts and full texts of the eleven articles should be stated.

Authors’ response: revised

3.2.3 The search strategy only provided for Medline and is not complete, since no terms for uvulectomy were described – the full search strategy for all databases is required.

Authors’ response: revised

3.2.4 Studies were considered low risk when they fit quality assessment checklists of 6 or above – please expand and clarify this statement and provide more detail describing how risk of bias was assessed.

Authors’ response: Revised

3.3 Outcome measurement and statistical analysis

3.3.1 Under outcome measurement, the statement, “all the procedures of cutting off a part of the uvula by traditional healers in the childhood age group (birth to under 15) in all African countries” appears to be missing a section – please edit and clarify.

Authors’ response: revised

3.3.2 What was the rationale for using the weighted inverse variance random-effects model?

Authors’ response: Variability Across Studies: In meta-analyses, individual studies often have different sample sizes, populations, methodologies, and contexts. A random-effects model assumes that there is some degree of variability between studies and that these differences contribute to the overall effect size. The model allows for the possibility that the true effect varies from study to study rather than assuming a single common effect (as a fixed-effects model would).

Weighting by Precision: The model uses the inverse variance method to assign weights to individual studies based on their precision (typically related to sample size and variability in the study results). Studies with larger sample sizes or lower variability receive higher weights because they provide more reliable estimates, whereas smaller or less precise studies are given less weight.

Generalizability of Results: By considering both within-study and between-study variability, the random-effects model produces more conservative and generalizable estimates of the overall effect size. This is particularly useful when pooling results from diverse studies, as it reflects the uncertainty in estimating the overall effect across different settings and populations.

Heterogeneity: The model is well-suited when there is significant heterogeneity (i.e., differences in effect sizes) between studies, as indicated by measures like I² statistics. A random-effects model adjusts for this variability, providing more robust and reliable pooled estimates when heterogeneity is present, which is especially important in contexts with diverse cultural and healthcare settings, such as studies on traditional uvulectomy practices in Africa.

3.3.3 Does “considering Ethiopia and other African countries” mean “comparing Ethiopia to other African countries”? Please clarify.

Authors’ response: yes, it is to compare Ethiopia and other countries.

3.3.4 What categories were used for synthesis and why? (see also 3.1.1)

Authors’ response: If I got the question, we categorized Ethiopian and other African countries

3.3.5 How were missing summary statistics handled ?

Authors’ response: Missing summary statistics were handled by contacting study authors for the required data. If unavailable, data imputation methods, such as estimation from available information (e.g., using reported confidence intervals, p-values, or medians), were applied to fill in gaps. So, as this is a review, we included the criteria of robust handling of primary studies to be included in the review.

3.3.6 What methods were used to tabulate or visually display results of individual studies and syntheses?

Authors’ response: Results of individual studies and syntheses were tabulated using summary tables that included study characteristics, prevalence, and odds ratios. Forest plots were used to visually display the pooled estimates and heterogeneity across studies.

3.3.7 What methods were used to assess risk of bias due to missing results in a synthesis (arising from reporting biases)?

Authors’ response: The risk of bias due to missing results from reporting biases was assessed using funnel plots and Egger's test to detect publication bias. Here we used egers test to see it objectively.

3.3.8 What p-values were used to interpret the publication bias assessment?

Authors’ response: In the table, Bias has a coefficient of 62.57 with a p-value of 0.098, which is above the typical threshold for statistical significance (0.05). However, it is approaching significance, suggesting a potential but not definitive indication of publication bias. The 95% confidence interval for Bias (from -14.23 to 139.37) is wide and includes zero, which means that there is still uncertainty about the presence of actual bias (Table 2).

In summary, while there is a hint of possible publication bias based on the bias coefficient, it is not statistically significant, and the wide confidence interval suggests that the evidence for publication bias is weak.

4. Results

4.1 Figure 1

4.1.1 The text in box, “The title and abstract do not fit” should be reworded more scientifically (eg “Title and/or abstract not relevant”)

4.1.2 The numbers of records per data base should be stated in the figure

Authors’ response: Revised

4.2 Characteristics of included studies

4.2.1 Only seven studies in Ethiopia were identified, compared to 19 in the review by Getachew et al. (International Journal of Pediatric Otorhinolaryngology 176 (2024) 111835), despite the same eligibility criteria – please explain the discrepancy to the reviewer and ensure that this is discussed in the discussion.

Authors’ response: this might be due to outcome variation and quality appraisal assessment checklist, we included all articles with 6 and above out of 9. We used articles at least with one associated factors and prevalence, and also the above mentioned author might use preprints not yet published. We only in

---

## [Decision Letter · Decision Letter 1]

20 Nov 2024

PONE-D-24-36948R1Prevalence of traditional uvulectomy and associated predictors among children in Africa: A systematic review and meta-analysisPLOS ONE

Dear Dr. Solomon,

Thank you for submitting your manuscript to PLOS ONE. After careful consideration, we feel that it has merit but does not fully meet PLOS ONE’s publication criteria as it currently stands. Therefore, we invite you to submit a revised version of the manuscript that addresses the points raised during the review process.

We look forward to receiving your revised manuscript.

Kind regards,

Kahsu Gebrekidan, Ph.D.

Academic Editor

PLOS ONE

Additional Editor Comments (if provided):

Dear Authors,

One of the reviewers suggest that you did not address all the previous comments.

Please address all the comments seriously before we proceed with further decision.

Revise the English seriously, preferably with someone who is native English speaker.

Reviewers' comments:

Reviewer's Responses to Questions

**Comments to the Author**

1. If the authors have adequately addressed your comments raised in a previous round of review and you feel that this manuscript is now acceptable for publication, you may indicate that here to bypass the “Comments to the Author” section, enter your conflict of interest statement in the “Confidential to Editor” section, and submit your "Accept" recommendation.

Reviewer #1: (No Response)

Reviewer #2: All comments have been addressed

Reviewer #3: (No Response)

2. Is the manuscript technically sound, and do the data support the conclusions?

Reviewer #1: Partly

Reviewer #2: Yes

Reviewer #3: Partly

3. Has the statistical analysis been performed appropriately and rigorously?

Reviewer #1: No

Reviewer #2: Yes

Reviewer #3: N/A

4. Have the authors made all data underlying the findings in their manuscript fully available?

Reviewer #1: Yes

Reviewer #2: Yes

Reviewer #3: (No Response)

5. Is the manuscript presented in an intelligible fashion and written in standard English?

Reviewer #1: No

Reviewer #2: Yes

Reviewer #3: No

6. Review Comments to the Author

Reviewer #1: A. Summary

The authors have stated in their reply that have addressed all the issues, however there are very few changes to the manuscript, many of the issues have not been addressed, and those that have been addressed have predominantly only been addressed in the response to reviewer and have not been added to the manuscript. In particular, the aims and objectives are still not clearly stated at the end of the background/introduction and the subheadings required by the PRISMA guidelines have still not been added to the abstract.

The authors have not numbered the lines in their manuscript, which is a journal requirement and there has been little change to the multiple grammatical errors, which are too numerous to list as part of this review.

I have listed the outstanding issues below, excluding all the grammatical errors, but considering the substantial and outstanding corrections required, I do not believe that this paper can be accepted.

B. Detailed issues

1. Abstract

1.1 The abstract is still missing multiple aspects which are required by PRISMA guidelines – the use of the subheadings listed in the PRISMA abstract check list is recommended, to ensure all aspects are included.

1.2 The following aspects must all still be addressed succinctly in the abstract.

1.2.1 The standard adhered to (PRISMA) should be stated.

1.2.2 The objectives need to be explicitly stated, describing the outcome (TCU)

1.2.3 The inclusion criteria need to be explicitly stated

1.2.4 The methods need to be briefly described.

1.2.5 The information sources/databases need to be stated

1.2.6 The risk of bias methods need to be stated

1.2.7 The characteristics of studies need to be stated

1.2.8 The limitations of the evidence included need to be stated

2. Introduction/Background

2.1 The statement: “Nevertheless, currently, in some parts of Sub-saharan Africa, as the majority of

people depend on traditional medicine for their primary healthcare demands” appears to be missing some words and is not understandable as it stands.

2.2 The line, “A systematic intervention strategy in the current health care delivery system in Africa using the pooled prevalence and predictors of TCU could be launched”, needs to be rephrased as an aim and specific objectives must be stated, using the word, “objectives” : a clear statement at the end of the introduction stating aims and specific objectives is required.

3. Methods

3.1 Add or clarify the following aspects:

3.1.1 The statement, “Subgroup analysis was performed by considering Ethiopia and other African countries, study settings, and sample size categories to overcome the inflation of the pooled effect from the inclusion of studies.” Should more precisely and clearly state the exact subgroups.

3.1.2 The start date stated in the author response must be included in the methods section of the manuscript.

3.2 Search strategy and quality assessment

3.2.1 The statement “The meta-analysis also included studies published in English and studies conducted only in African countries.” Needs to be rephrased to more clearly state that ONLY studies published in English and in Africa were included.

3.2.2 The last line of the paragraph with title, “Study selection: states, “Two independent authors subsequently reviewed the abstracts and full texts of eleven articles” – the initials of those authors must be stated.

3.2.3 The search strategy was only provided for Medline and is not complete, since no terms for uvulectomy were described. The full search strategy for all databases is required by PRISMA standards.

3.2.4 There is insufficient detail in the main text of the manuscript summarising how risk of bias was assessed – the reader should not have to refer to supplementary files.

3.3 Outcome measurement and statistical analysis

3.3.1 Under outcome measurement, the statement, “all the procedures of cutting off a part of the uvula by traditional healers in the childhood age group (birth to under 15) in all African countries” appears to be missing a section – this has not been corrected as requested - please edit and clarify.

3.3.2 Please provide a summary of the rationale for using the weighted inverse variance random-effects model in the text of the manuscript.

3.3.3 The statement, “considering Ethiopia and other African countries” should be edited in the text to, “comparing Ethiopia to other African countries”

3.3.4 Please state, in the manuscript, the rationale for grouping Ethiopia separately to other African countries.

3.3.5 Please provide a brief clear statement in the manuscript in the methods, stating which categories were used for synthesis and why?

3.3.6 Please briefly describe in the manuscript, how were missing summary statistics were handled.

3.3.7 Please succinctly describe in the manuscript, what methods were used to tabulate or visually display results of individual studies and syntheses.

3.3.8 Please succinctly describe in the manuscript what methods were used to assess risk of bias due to missing results in a synthesis (not the same as publication bias)

3.3.9 Please describe in the manuscript, what p-values were used to interpret the publication bias assessment.

4. Results

4.1 Figure 1

The revised figure 1 was not included – please supply it to determine if the below amendments were done.

4.1.1 The text in box, “The title and abstract do not fit” should be reworded more scientifically (eg “Title and/or abstract not relevant”)

4.1.2 The numbers of records per data base should be stated in the figure

4.2 Characteristics of included studies

4.2.1 Only seven studies in Ethiopia were identified, compared to 19 in the review by Getachew et al. (International Journal of Pediatric Otorhinolaryngology 176 (2024) 111835), despite the same eligibility criteria – please explain the discrepancy to the reviewer and ensure that this is discussed in the discussion. The authors need to address the above question factually. In their response to the reviewer, they speculate what the possible reasons may be, however that is not acceptable – they need to thoroughly read the paper and clearly and factually explain in the manuscript, the reasons for the differences, in the discussion section of their manuscript.

4.2.2 The table listing the characteristics of included studies should be referenced at the beginning of the paragraph, not at the end.

4.2.3 In the manuscript, the term, “risk of bias” should be used in preference to “risk”.

4.2.4 The term “included” should be used rather than “considered”, since all were included.

4.2.5 Please add the word, “all” to the statement (as underlined ) , “none of the included studies were of poor quality and all had a quality score ≥6 out of 9 criteria”

4.2.6 Please edit “JBI quality appraisal checklist,” to “JBI risk of bias quality appraisal checklist”

4.2.7 The 9 quality criteria should be stated, and the scores should be included separately in a graphic. Supplementary file 2 should be included in the mian text of the manuscript as a table, with “yes” “no” “low risk” presented as graphics, rather than text.

4.2.8 Please add a table in the manuscript with details of the ten excluded studies which, “did not report the outcome of interest”.

4.3 Meta-analysis

4.3.1 It is more intuitive to the reader to first read about the pooled incidence of TCU, subgroup analysis and predictors , before describing the Egger’s test, so it is clear what outcomes it refers to.

4.3.2 Egger’s test assesses publication bias, not “absence of publication bias” – the text should briefly explain how the parameters in table show no publication bias.

4.3.3 The title of table 2 could rather be stated as, “Egger’s test for publication bias in the effect sizes of prevalence and predictors of TCU in Africa.”

4.3.4 What is meant by, “mothers with a lack of information and who were perceived as

uvula causes”? Please clarify in the manuscript.

4.4 Subgroup analysis

4.4.1 Please add more more detail in the manuscript describing exactly which African countries, types of communities and types of health facilities were grouped.

4.4.2 The interpretation of the results with statements such as, “This implies that nearly half of the children would not seek medical care after they had received TCU”, should be moved from the results section to the discussion.

4.4.3 Table 3 needs a more detailed title explaining what it represents. Needs footnotes with abbreviations and explanations of the subgroups – eg, which other African countries, what does community-based mean? What does health facilities mean?

4.4.4 The associated forest plots for the subgroup analyses should be included in the text (these are different to figures 2 – 4, and could not be found by the reviewer)

4.5 Predictors of TCU

4.5.1 Please correct the grammar of the statement, “nearly fourfold more patients with uvulitis not cured by medical treatment not cured by medical treatment than did their counterparts”

4.5.2 Please correct the grammar/syntax of the statement, in the last line of the results, “AOR=4.31; 95% CI=2.17-6[8,9]/>, respectively).”

5. Discussion

5. 1 The discussion should begin with statement referring to what was described in the study, rather than repeating the aim.

5.2 Please discuss the results for the Ethiopian studies in detailed comparison to the systematic review of TCU in Ethiopia by Getachew et al.- see 4.2.1

5.3 The predictors of TCU were rural residency and a history of previous practice – please discuss in the text of the manuscript, why maternal education was not included, since this was identified in the more comprehensive systematic review of TCU in Ethiopia by Getachew et al.?

5.4 The statement “the most common utilizers of traditional medicine were found in India, China, and Africa”, appears to refer to the current study, which was limited to Africa – please reword the statement in your manuscript to clarify which studies it refers to.

5.5 Please add a paragraph at the end of the discussion in the manuscript describing the strengths and limitations of your study.

5.6 Please add a paragraph in the conclusion of your manuscript, clearly, precisely and succinctly explaining how the findings of your study can be used to develop a project addressing TCU?

Reviewer #2: (No Response)

Reviewer #3: Thank you for the opportunity to rereview the text.

The authors attempted to address some comments provided by the reviewers.

I remain of the opinion that the text needs to be screened in depth for grammar, punctuation, tenses and complete sentences and use of correct phrases. Reviewer #3 made several comments/suggestions to improve this, which are not adequately addressed, but which might have improved clarity for the reader.

The text still implies, in part, that mothers/parents either practice TCU themselves or receive TCU themselves, instead of their children.

Search criteria and results criteria for platforms searched are incongruent. Also, no results were highlighted for Science Direct. If necessary, a PRISMA search diagram may be included in the text (as opposed to a supplement) to improve the section for search/results of literature.

Reviewer #2 requested search results be included into the Abstract.

Please check consistency of spelling for "sub-Saharan"

Thank you again.

7. PLOS authors have the option to publish the peer review history of their article (what does this mean? ). If published, this will include your full peer review and any attached files.

**Do you want your identity to be public for this peer review?** For information about this choice, including consent withdrawal, please see our Privacy Policy .

Reviewer #1: No

Reviewer #2: **Yes: ** Seid Muhumed Abdilaahi, Departments of Pediatrics and Child Health Nursing, Institute of Health Science, Jigjiga University, Jigjiga, Ethiopia

Reviewer #3: **Yes: ** Beatrix Callard

---

## [Author Response · Author response to Decision Letter 1]

23 Nov 2024

Reviewer #1: A. Summary

The authors have stated in their reply that have addressed all the issues, however there are very few changes to the manuscript, many of the issues have not been addressed, and those that have been addressed have predominantly only been addressed in the response to reviewer and have not been added to the manuscript. In particular, the aims and objectives are still not clearly stated at the end of the background/introduction and the subheadings required by the PRISMA guidelines have still not been added to the abstract.

Authors’ response: it is added as ‘This study adhered to the PRISMA (Preferred Reporting Items for Systematic Reviews and Meta-Analyses) guidelines to ensure transparency and reproducibility’ in the abstract and the objective also modified in the introduction section of the manuscript as ‘This study aims to estimate the pooled prevalence of traditional childhood uvulectomy (TCU) practices in Africa’.

The authors have not numbered the lines in their manuscript, which is a journal requirement and there has been little change to the multiple grammatical errors, which are too numerous to list as part of this review.

Authors’ response: revised by numbering lines and language editing.

I have listed the outstanding issues below, excluding all the grammatical errors, but considering the substantial and outstanding corrections required, I do not believe that this paper can be accepted.

Authors’ response: We revised thoroughly as much as possible and please reconsider your decision this paper to be accepted. We devoted our time and effort as much as all authors concerned.

B. Detailed issues

1. Abstract

1.1 The abstract is still missing multiple aspects which are required by PRISMA guidelines – the use of the subheadings listed in the PRISMA abstract check list is recommended, to ensure all aspects are included.

Authors’ response: revised

1.2 The following aspects must all still be addressed succinctly in the abstract.

1.2.1 The standard adhered to (PRISMA) should be stated.

1.2.2 The objectives need to be explicitly stated, describing the outcome (TCU)

1.2.3 The inclusion criteria need to be explicitly stated

1.2.4 The methods need to be briefly described.

1.2.5 The information sources/databases need to be stated

1.2.6 The risk of bias methods need to be stated

1.2.7 The characteristics of studies need to be stated

1.2.8 The limitations of the evidence included need to be stated

Authors’ response: revised by using it as per comments given

2. Introduction/Background

2.1 The statement: “Nevertheless, currently, in some parts of Sub-saharan Africa, as the majority of people depend on traditional medicine for their primary healthcare demands” appears to be missing some words and is not understandable as it stands.

Authors’ response: I did not get where that unclear statement is found. But, we tried to clarify each paragraph again in this revision.

2.2 The line, “A systematic intervention strategy in the current health care delivery system in Africa using the pooled prevalence and predictors of TCU could be launched”, needs to be rephrased as an aim and specific objectives must be stated, using the word, “objectives” : a clear statement at the end of the introduction stating aims and specific objectives is required.

Authors’ response: revised as ‘The objective of this review and meta-analysis was estimating pooled prevalence and predictors of traditional childhood uvulectomy (TCU) practices in Africa.’

3. Methods

3.1 Add or clarify the following aspects:

3.1.1 The statement, “Subgroup analysis was performed by considering Ethiopia and other African countries, study settings, and sample size categories to overcome the inflation of the pooled effect from the inclusion of studies.” Should more precisely and clearly state the exact subgroups.

Authors’ response: Subgroup analysis was conducted based on the following factors: country (Ethiopia, which had a higher concentration of primary studies, and other African countries), study setting (community-based or health facility-based), and sample size (categorized as less than 500 or 500 and above).

3.1.2 The start date stated in the author response must be included in the methods section of the manuscript.

Authors’ response: add on the manuscript (the start date).

3.2 Search strategy and quality assessment

3.2.1 The statement “The meta-analysis also included studies published in English and studies conducted only in African countries.” Needs to be rephrased to more clearly state that ONLY studies published in English and in Africa were included.

Authors’ response: corrected

3.2.2 The last line of the paragraph with title, “Study selection: states, “Two independent authors subsequently reviewed the abstracts and full texts of eleven articles” – the initials of those authors must be stated.

Authors’ response: revised as ‘Two independent authors (TMA and AK) reviewed the abstracts and full texts of eleven articles published between 2010 and 2024.’

3.2.3 The search strategy was only provided for Medline and is not complete, since no terms for uvulectomy were described. The full search strategy for all databases is required by PRISMA standards.

Authors’ response: we used similar the search strategies to all database by using different Boolean operators mentioned in the manuscript.

3.2.4 There is insufficient detail in the main text of the manuscript summarising how risk of bias was assessed – the reader should not have to refer to supplementary files.

Authors’ response: revised

3.3 Outcome measurement and statistical analysis

3.3.1 Under outcome measurement, the statement, “all the procedures of cutting off a part of the uvula by traditional healers in the childhood age group (birth to under 15) in all African countries” appears to be missing a section – this has not been corrected as requested - please edit and clarify.

Authors’ response: revised as ‘’ Traditional Childhood Uvulectomy (TCU) is a cultural practice in which a portion of the uvula (the small, fleshy extension at the back of the soft palate) is surgically removed or altered by traditional healers or unqualified practitioners, typically without medical supervision on under 15 age group in Africa.’’

3.3.2 Please provide a summary of the rationale for using the weighted inverse variance random-effects model in the text of the manuscript.

Authors’ response: add a reason as ‘This model was selected because it is better suited to handle the variability between studies, providing a more nuanced and reliable estimate of the pooled prevalence of TCU and its predictors across Africa, compared to a fixed-effects model’.

3.3.3 The statement, “considering Ethiopia and other African countries” should be edited in the text to, “comparing Ethiopia to other African countries”

Authors’ response: revised

3.3.4 Please state, in the manuscript, the rationale for grouping Ethiopia separately to other African countries.

Authors’ response: as there is a number of studies in Ethiopia which is comparable to other countries combined.

3.3.5 Please provide a brief clear statement in the manuscript in the methods, stating which categories were used for synthesis and why?

Authors’ response: Subgroup analysis was conducted based on the following factors: country (Ethiopia, which had a higher concentration of primary studies, and other African countries), study setting (community-based or health facility-based), and sample size (categorized as less than 500 or 500 and above). As this is the possible category to have a maximum possibility to balance the study participants in primary studies, we used the aforementioned category.

3.3.6 Please briefly describe in the manuscript, how were missing summary statistics were handled.

3.3.7 Please succinctly describe in the manuscript, what methods were used to tabulate or visually display results of individual studies and syntheses.

Authors’ response: tables and figures

3.3.8 Please succinctly describe in the manuscript what methods were used to assess risk of bias due to missing results in a synthesis (not the same as publication bias)

Authors’ response: we tried to show it by publication bias-Eger’s test. And we took care of searching of literatures robustly and in addition doing sensitivity analysis to show each studies effect on the overall pooled prevalence. I feel that I did not answer your question, would you recommend me how to incorporate it.

3.3.9 Please describe in the manuscript, what p-values were used to interpret the publication bias assessment.

Authors’ response: Publication bias assessed by Eger’s’ test. The absence of publication bias was assessed using Egger’s regression test. A significant result (P<0.05) suggests potential publication bias, while a non-significant result (P>0.05) indicates no evidence of bias. In this analysis, Egger’s test yielded p = 0.098, indicating no significant evidence of publication bias.

4. Results

4.1 Figure 1

The revised figure 1 was not included – please supply it to determine if the below amendments were done.

4.1.1 The text in box, “The title and abstract do not fit” should be reworded more scientifically (eg “Title and/or abstract not relevant”)

4.1.2 The numbers of records per data base should be stated in the figure

Authors’ response: is that mean the number of articles found, is so, it is considered

4.2 Characteristics of included studies

4.2.1 Only seven studies in Ethiopia were identified, compared to 19 in the review by Getachew et al. (International Journal of Pediatric Otorhinolaryngology 176 (2024) 111835), despite the same eligibility criteria – please explain the discrepancy to the reviewer and ensure that this is discussed in the discussion. The authors need to address the above question factually. In their response to the reviewer, they speculate what the possible reasons may be, however that is not acceptable – they need to thoroughly read the paper and clearly and factually explain in the manuscript, the reasons for the differences, in the discussion section of their manuscript.

Authors’ response: We reviewed the paper but did not find a definitive answer to the question. As previously mentioned there is considerable variability in both objectives and timing across studies=the objective for Getachew et al 2024 study is for complication burden secondary to uvulectomy not the pooled prevalence. (https://www.cell.com/heliyon/fulltext/S2405-8440(24)15009-8?uuid=uuid%3Ac15fd92c-c193-4caa-b8c3-ed7d45f37f0c ). what is more confused about it is in other journal it seems pooled prevalence (file:///C:/Users/HAB-TECH/Downloads/Uvlectomy%202.pdf). While we excluded qualitative studies, the study by Getachew et al. incorporated mixed-method/title studies. For example, it included research assessing knowledge of neonatal care, where uvulectomy was mentioned as one aspect but was not the primary focus. Getachew et al. also included articles exploring knowledge and attitudes, which often relied on qualitative methods to investigate reasons for uvulectomy.

In contrast, our review primarily focused on the practice of uvulectomy and its associated factors. As a result, we excluded several studies included by Getachew et al. For instance, Hadush A., 2016 (titled “Assessment of Knowledge and Practice of Neonatal Care among Postnatal Mothers Attending Ayder and Mekelle Hospital in Mekelle, Tigray, Ethiopia 2013”) was excluded from our review because it primarily addressed neonatal care rather than uvulectomy.

Additional factors explain the differences between our review and that of Getachew et al. First, we included only studies scoring 6 or above out of 9 on our quality appraisal checklist. Second, we focused on articles that reported at least one associated factor or the prevalence of uvulectomy. Third, Getachew et al. may have included preprints or unpublished studies, whereas we restricted our review to published articles to ensure quality assurance.

Finally, differences in population scope may also contribute to discrepancies. Our review specifically targeted childhood uvulectomy (ages birth to 15 years), whereas Getachew et al.’s inclusion criteria may have encompassed broader populations or varying age groups.

4.2.2 The table listing the characteristics of included studies should be referenced at the beginning of the paragraph, not at the end.

Authors’ response: corrected

4.2.3 In the manuscript, the term, “risk of bias” should be used in preference to “risk”.

Authors’ response: revised

4.2.4 The term “included” should be used rather than “considered”, since all were included.

Authors’ response: considered

4.2.5 Please add the word, “all” to the statement (as underlined ) , “none of the included studies were of poor quality and all had a quality score ≥6 out of 9 criteria”

Authors’ response: revised

4.2.6 Please edit “JBI quality appraisal checklist,” to “JBI risk of bias quality appraisal checklist”

4.2.7 The 9 quality criteria should be stated, and the scores should be included separately in a graphic. Supplementary file 2 should be included in the mian text of the manuscript as a table, with “yes” “no” “low risk” presented as graphics, rather than text.

Authors’ response: as it is checklist, it makes the reader unpleasant I think. If it is must, I will do.

Table 2: Quality Assessment of the Included Studies Using JBI Risk of Bias Assessment checklist

Criteria Score Quality

Author Was The Sample Frame Appropriate? Was Sampling Appropriate? Was The Sample Size Adequate? Were The Study Subjects And The Setting Described In Detail? Was The Data Analysis Conducted With Sufficient Coverage Of The Identified Sample? Were Valid Methods Used For The Identification Of The Condition? Was The Condition Measured In A Standard, Reliable Way For All Participants? Was There Appropriate Statistical Analysis? Was The Response Rate Adequate, And If Not, Was The Low Response Rate Managed Appropriately?

(Farouk et al., (2023)

Yes Yes Yes Yes Yes Yes Yes Yes Yes 9 Low risk

(Kambale et al., 2018)

Yes Yes No Yes No Yes Yes Yes No 6 Low risk

(Adoga, 2011)

Yes Yes Yes No Yes Yes No Yes Yes 7 Low risk

(Oluwatosin et al., 2016)

Yes Yes Yes Yes Yes Yes Yes Yes Yes 9 Low risk

Djakounda, 1994) Yes Yes Yes Yes Not Clear Yes Yes Yes Yes 8 Low risk

(Gebrekirstos et al., 2014)

Yes Yes Yes Not Clear Yes Yes Yes Yes Yes 8 Low risk

(Mitke, 2010)

Yes Yes Yes Yes Yes Yes Yes Yes Yes 9 Low risk

(Kebede et al., 2017)

Yes Yes Yes Yes Yes Yes No Yes No 6 Low risk

(Yirdaw et al., 2022)

Yes Yes No Yes Yes No No Yes Yes 6 Low risk

(Gebrekirstos et al., 2014)

Yes Yes Yes Yes No Yes Yes No No 6 Low risk

(Bayih et al., 2020)

Yes Yes Yes Yes Yes Yes Yes Yes Yes 9

4.2.8 Please add a table in the manuscript with details of the ten excluded studies which “did not report the outcome of interest”.

Authors’ response: added excluded studies of this manuscript as Table 2.

4.3 Meta-analysis

4.3.1 It is more intuitive to the reader to first read about the pooled incidence of TCU, subgroup analysis and predictors , before describing the Egger’s test, so it is clear what outcomes it refers to.

4.3.2 Egger’s test assesses publication bias, not “absence of publication bias” – the text should briefly explain how the parameters in table show no publication bias.

Authors’ response: revised

4.3.3 The title of table 2 could rather be stated as, “Egger’s test for publication bias in the effect sizes of prevalence and predictors of TCU in Africa.”

Authors’ response: revised as per comments lines 17-22.

4.3.4 What is meant by, “mothers with a lack of information and who were perceived as

uvula causes”? Please clarify in the manuscript.

Authors’ response: revised as’’ In a single study by Yirdaw et al., mothers who lacked information about uvulectomy and those who perceived the uvula as a cause of infection were significantly more likely to practice traditional childhood uvulectomy (TCU) compared to those who were informed about its complications’’.

4.4 Subgroup analysis

4.4.1 Please add more more detail in the manuscript describing exactly which African countries, types of communities and types of health facilities were grouped.

Authors’ response: revised in

---

## [Decision Letter · Decision Letter 2]

29 Nov 2024

PONE-D-24-36948R2Pooled prevalence and associated factors of traditional uvulectomy among children in Africa: A systematic review and meta-analysisPLOS ONE

Dear Dr. Demis,

Thank you for submitting your manuscript to PLOS ONE. After careful consideration, we feel that it has merit but does not fully meet PLOS ONE’s publication criteria as it currently stands. Therefore, we invite you to submit a revised version of the manuscript that addresses the points raised during the review process.

We look forward to receiving your revised manuscript.

Kind regards,

Kahsu Gebrekidan, Ph.D.

Academic Editor

PLOS ONE

Journal Requirements:

Reviewers' comments:

Reviewer's Responses to Questions

**Comments to the Author**

1. If the authors have adequately addressed your comments raised in a previous round of review and you feel that this manuscript is now acceptable for publication, you may indicate that here to bypass the “Comments to the Author” section, enter your conflict of interest statement in the “Confidential to Editor” section, and submit your "Accept" recommendation.

Reviewer #3: (No Response)

2. Is the manuscript technically sound, and do the data support the conclusions?

Reviewer #3: Partly

3. Has the statistical analysis been performed appropriately and rigorously?

Reviewer #3: Yes

4. Have the authors made all data underlying the findings in their manuscript fully available?

Reviewer #3: Yes

5. Is the manuscript presented in an intelligible fashion and written in standard English?

Reviewer #3: Yes

6. Review Comments to the Author

Reviewer #3: Congratulations. The grammar, sentence structure and flow of the text has greatly improved and thus provides clearer insight. Minor comments and suggestions are attached.

My major concern is the lack of clarity (perhaps on my part) about the agents involved in the practice of TCU. To my understanding, there are three agents in this transaction.

1. The traditional healer – the one who performs the procedure and who practices TCU

2. The mother/parent – the one who seeks/requests this procedure for their child

3. The child who receives or undergoes this procedure (I assume not by choice)

The language in the text implies mothers practice and undergo this procedure. The role of the three agents outlined above are not clearly delineated in the text and confuse me as to who is responsible for doing the cutting (practice) and who had what done to them.

Are the authors saying that mothers who themselves had TCU done to them when they were children are more likely to have their own children undergo TCU? Or are the authors saying that mothers who have older children, who have undergone TCU with or without complications) are more likely to seek TCU for their younger children? The difference is subtle but important for interpretation of ongoing use and provision of traditional health care services in general.

Comments/Suggestions below:

Line number

Abstract:

29 ...associated of TCU… – there is a word missing

47 “perform”- suggest “seek” or “request”

52 …in Africa… – over inflating of results. Suggest to amend to read “…in some African countries…”

Text

71 double period

88 incomplete sentence – suggest, remove the word “as”

90 unsuportedand – add space; remove the word “possibly” as you are confirming the danger of this practice later in the paragraph.

95 …during the same session… - suggest replace “session” with “procedure” – adds clarity.

135, 138, 139 Change oorder of words to align with P E C O (Context before outcome)

143 & 165 associated factors or “predictors”?

187 & 188 use past tense (were,was instead of are, is)

212 Did Science Direct yield a zero result search? This may be worth mentioning.

Page 12 restarts line numbers as 1.

1 risk of what? Specify if this relates to risk of bias.

63 predictors of TCU …as rural residents and had a previous history to TCU… - sentence is slightly confusing – I am still unclear if “previous history” refers to the mother who herself has undergone the procedure of TCU or if she has had a previous child whom underwent the procedure. This aspect needs clarity!

64 …mothers were more likely to undergo… - again, this implies the women themselves are having the uvulas removed – and not their children.

68 &73 &76 & 86 as with line 63… -did the mothers themselves have the TCU or did they seek this for an older child. Also, Mothers do not “practice” this procedure – they either seek or request this for their children… Please clarify this across the document.

78 patients with uvulitis – is this in the adult population or the paediatric population – this is unclear.

83 children practice – children cannot do this procedure themselves (practice), they can undergo or receive this procedure.

7. PLOS authors have the option to publish the peer review history of their article (what does this mean? ). If published, this will include your full peer review and any attached files.

**Do you want your identity to be public for this peer review?** For information about this choice, including consent withdrawal, please see our Privacy Policy .

Reviewer #3: **Yes: ** Beatrix Callard

---

## [Author Response · Author response to Decision Letter 2]

1 Dec 2024

Journal Requirements:

Authors’ response: we checked for retracted articles and no article was found with retracted note.

Reviewers' comments:

Reviewer's Responses to Questions

Comments to the Author

1. If the authors have adequately addressed your comments raised in a previous round of review and you feel that this manuscript is now acceptable for publication, you may indicate that here to by pass the “Comments to the Author” section, enter your conflict of interest statement in the “Confidential to Editor” section, and submit your "Accept" recommendation.

Reviewer #3: (No Response)

2. Is the manuscript technically sound, and do the data support the conclusions?

Reviewer #3: Partly

3. Has the statistical analysis been performed appropriately and rigorously?

Reviewer #3: Yes

4. Have the authors made all data underlying the findings in their manuscript fully available?

Reviewer #3: Yes

5. Is the manuscript presented in an intelligible fashion and written in standard English?

Reviewer #3: Yes

6. Review Comments to the Author

Reviewer #3: Congratulations. The grammar, sentence structure and flow of the text has greatly improved and thus provides clearer insight. Minor comments and suggestions are attached.

My major concern is the lack of clarity (perhaps on my part) about the agents involved in the practice of TCU. To my understanding, there are three agents in this transaction.

1. The traditional healer – the one who performs the procedure and who practices TCU

2. The mother/parent – the one who seeks/requests this procedure for their child

3. The child who receives or undergoes this procedure (I assume not by choice)

The language in the text implies mothers practice and undergo this procedure. The role of the three agents outlined above are not clearly delineated in the text and confuse me as to who is responsible for doing the cutting (practice) and who had what done to them.

Are the authors saying that mothers who themselves had TCU done to them when they were children are more likely to have their own children undergo TCU? Or are the authors saying that mothers who have older children, who have undergone TCU with or without complications) are more likely to seek TCU for their younger children? The difference is subtle but important for interpretation of ongoing use and provision of traditional health care services in general.

Authors’ response: I sincerely appreciate the comments and acknowledge their importance. They raise very crucial points.

To clarify, we are referring to mothers whose children underwent a traditional uvulectomy. The mothers sought out traditional healers to perform the procedure on their children, but the mothers themselves did not perform it.

Regarding the reference to a previous history of uvulectomy, this indicates mothers who have one or more children who have already undergone the procedure.

Thank you for bringing up these points, as they help refine the understanding and presentation of the context.

Comments/Suggestions below:

Line number

Abstract:

29 ...associated of TCU… – there is a word missing

47 “perform”- suggest “seek” or “request”

52 …in Africa… – over inflating of results. Suggest to amend to read “…in some African countries…”

Authors’ response: revised as per comments.

Text

71 double period

88 incomplete sentence – suggest, remove the word “as”

90 unsuportedand – add space; remove the word “possibly” as you are confirming the danger of this practice later in the paragraph.

95 …during the same session… - suggest replace “session” with “procedure” – adds clarity.

135, 138, 139 Change oorder of words to align with P E C O (Context before outcome)

143 & 165 associated factors or “predictors”?

Authors’ response: revised as per comments.

187 & 188 use past tense (were,was instead of are, is)

212 Did Science Direct yield a zero result search? This may be worth mentioning.

Page 12 restarts line numbers as 1.

Authors’ response: Yes, there were no studies included for analysis from ScienceDirect, but there were five excluded articles at screening stage found in the search. At that stage, I considered only the databases that contributed to the final articles analyzed in this review.

1 risk of what? Specify if this relates to risk of bias.

Authors’ response: Risk of bias.

63 predictors of TCU …as rural residents and had a previous history to TCU… - sentence is slightly confusing – I am still unclear if “previous history” refers to the mother who herself has undergone the procedure of TCU or if she has had a previous child whom underwent the procedure. This aspect needs clarity!

Authors’ response: revised as follows:

‘’Mothers residing in rural areas were 2.45 times more likely to have a child experienced TCU compared to those in urban areas’’

‘’Mothers with a history of having a previous child who experienced TCU were 8.44 times more likely to seek the procedure for their other children compared to mothers without such a history.’

64 …mothers were more likely to undergo… - again, this implies the women themselves are having the uvulas removed – and not their children.

Authors’ response: revised as: By pooling studies, rural resident mothers were 2.45 times more likely to have a child who underwent traditional childhood uvulectomy (TCU) compared to mothers living in urban areas.

68 &73 &76 & 86 as with line 63… -did the mothers themselves have the TCU or did they seek this for an older child. Also, Mothers do not “practice” this procedure – they either seek or request this for their children… Please clarify this across the document.

Authors’ response: Revised

78 patients with uvulitis – is this in the adult population or the paediatric population – this is unclear.

83 children practice – children cannot do this procedure themselves (practice), they can undergo or receive this procedure.

Authors’ response: revised as ‘According to maternal attitudes, children with uvulitis (inflammation and swelling of the uvula) that did not respond to medical treatment were nearly four times more likely to undergo traditional uvulectomy compared to their counterparts’.

---

## [Decision Letter · Decision Letter 3]

13 Dec 2024

PONE-D-24-36948R3Pooled prevalence and associated factors of traditional uvulectomy among children in Africa: A systematic review and meta-analysisPLOS ONE

Dear Dr. Demis,

Thank you for submitting your manuscript to PLOS ONE. After careful consideration, we feel that it has merit but does not fully meet PLOS ONE’s publication criteria as it currently stands. Therefore, we invite you to submit a revised version of the manuscript that addresses the points raised during the review process.

Please go over your manuscript again and correct any possible language issues. Please note that PLOS ONE does not provide edit proof, any language concerns should be addressed before publication.

We look forward to receiving your revised manuscript.

Kind regards,

Kahsu Gebrekidan, Ph.D.

Academic Editor

PLOS ONE

Journal Requirements:

Reviewers' comments:

Reviewer's Responses to Questions

**Comments to the Author**

1. If the authors have adequately addressed your comments raised in a previous round of review and you feel that this manuscript is now acceptable for publication, you may indicate that here to bypass the “Comments to the Author” section, enter your conflict of interest statement in the “Confidential to Editor” section, and submit your "Accept" recommendation.

Reviewer #3: (No Response)

2. Is the manuscript technically sound, and do the data support the conclusions?

Reviewer #3: Yes

3. Has the statistical analysis been performed appropriately and rigorously?

Reviewer #3: N/A

4. Have the authors made all data underlying the findings in their manuscript fully available?

Reviewer #3: Yes

5. Is the manuscript presented in an intelligible fashion and written in standard English?

Reviewer #3: Yes

6. Review Comments to the Author

Reviewer #3: Minor edits.

Line 188 - change 'are' to 'were'

Line 62 - ...mothers with older children who had a previous history...

Line 67 change 'underwent' to 'undergo'

Line 83 - replace 'to have a child with TCU' with 'to receive TCU'

7. PLOS authors have the option to publish the peer review history of their article (what does this mean? ). If published, this will include your full peer review and any attached files.

**Do you want your identity to be public for this peer review?** For information about this choice, including consent withdrawal, please see our Privacy Policy .

Reviewer #3: **Yes: ** Beatrix Callard

---

## [Author Response · Author response to Decision Letter 3]

13 Dec 2024

We sincerely appreciate the reviewers’ valuable comments and acknowledge their importance in strengthening our manuscript. Below are our responses to the specific points raised:

Reviewer #3: Minor edits

1. Line 188: Change 'are' to 'were.'

Authors’ response: The correction has been made by replacing 'are' with 'were.'

2. Line 62: "...mothers with older children who had a previous history..."

Authors’ response: This section has been revised as per the comment for improved accuracy.

3. Line 67: Change 'underwent' to 'undergo.'

Authors’ response: The change has been implemented, and 'underwent' has been replaced with 'undergo.'

4. Line 83: Replace 'to have a child with TCU' with 'to receive TCU.'

Authors’ response: The sentence has been revised as suggested.

---

## [Editor Report · Decision Letter 4]

17 Dec 2024

Pooled prevalence and associated factors of traditional uvulectomy among children in Africa: A systematic review and meta-analysis

PONE-D-24-36948R4

Dear Mr. Solomon,

We’re pleased to inform you that your manuscript has been judged scientifically suitable for publication and will be formally accepted for publication once it meets all outstanding technical requirements.

Kind regards,

Kahsu Gebrekidan, Ph.D.

Academic Editor

PLOS ONE
---

## [Editor Report · Acceptance letter]

PONE-D-24-36948R4

PLOS ONE

Dear Dr. Demis,

I'm pleased to inform you that your manuscript has been deemed suitable for publication in PLOS ONE. Congratulations! Your manuscript is now being handed over to our production team.

Kind regards,

on behalf of

Dr. Kahsu Gebrekidan

Academic Editor

PLOS ONE